# Boosting with Tempered Exponential Measures

**Richard Nock**
Google Research
richardnock@google.com

**Ehsan Amid**
Google DeepMind
eamid@google.com

**Manfred K. Warmuth**
Google Research
manfred@google.com

## Abstract

One of the most popular ML algorithms, ADABOOST, can be derived from the dual of a relative entropy minimization problem subject to the fact that the positive weights on the examples sum to one. Essentially, harder examples receive higher probabilities. We generalize this setup to the recently introduced *tempered exponential measure*s (TEMs) where normalization is enforced on a specific power of the measure and not the measure itself. TEMs are indexed by a parameter $t$ and generalize exponential families ($t = 1$). Our algorithm, $t$-ADABOOST, recovers ADABOOST as a special case ($t = 1$). We show that $t$-ADABOOST retains ADABOOST's celebrated exponential convergence rate on margins when $t \in [0, 1)$ while allowing a slight improvement of the rate's hidden constant compared to $t = 1$. $t$-ADABOOST partially computes on a generalization of classical arithmetic over the reals and brings notable properties like guaranteed bounded leveraging coefficients for $t \in [0, 1)$. From the loss that $t$-ADABOOST minimizes (a generalization of the exponential loss), we show how to derive a new family of *tempered* losses for the induction of domain-partitioning classifiers like decision trees. Crucially, strict properness is ensured for all while their boosting rates span the full known spectrum of boosting rates. Experiments using $t$-ADABOOST+trees display that significant leverage can be achieved by tuning $t$.

## 1 Introduction

ADABOOST is one of the most popular ML algorithms [8, 30]. It efficiently aggregates weak hypotheses into a highly accurate linear combination [10]. The common motivations of boosting algorithms focus on choosing good linear weights (the leveraging coefficients) for combining the weak hypotheses. A dual view of boosting highlights the dual parameters, which are the weights on the examples. These weights define a distribution, and ADABOOST can be viewed as minimizing a relative entropy to the last distribution subject to a linear constraint introduced by the current hypothesis [12]. For this reason (more in § 2), ADABOOST's weights define an exponential family.

**In this paper**, we go beyond weighing the examples with a discrete exponential family distribution, relaxing the constraint that the total mass be unit but instead requiring it for the measure's $1/(2-t)$'th power, where $t$ is a temperature parameter. Such measures, called *tempered exponential measures* (TEMs), have been recently introduced [4]. Here we apply the discrete version of these TEMs for deriving a novel boosting algorithm called $t$-ADABOOST. Again the measures are solutions to a relative entropy minimization problem, but the relative entropy is built from Tsallis entropy and "tempered" by a parameter $t$. As $t \to 1$ TEMs become standard exponential family distributions and our new algorithm merges into ADABOOST. As much as ADABOOST minimizes the exponential loss, $t$-ADABOOST minimizes a generalization of this loss we denote as the *tempered exponential loss*.

TEMs were introduced in the context of clustering, where they were shown to improve the robustness to outliers of clustering's population minimizers [4]. They have also been shown to bring low-level sparsity features to optimal transport [3]. Boosting is a high-precision machinery: ADABOOST is known to achieve near-optimal boosting rates under the weak learning assumption [1], but it has

37th Conference on Neural Information Processing Systems (NeurIPS 2023).

long been known that numerical issues can derail it, in particular, because of the unbounded weight update rule [14]. So the question of what the TEM setting can bring for boosting is of primordial importance. As we show, $t$-ADABOOST can suffer no rate setback as boosting's exponential rate of convergence on *margins* can be preserved for all $t \in [0, 1)$. Several interesting features emerge: the weight update becomes bounded, margin optimization can be *tuned* with $t$ to focus on examples with very low margin and besides linear separators, the algorithm can also learn *progressively* clipped models[1]. Finally, the weight update makes appear a new regime whereby weights can "switch off and on": an example's weight can become zero if too well classified by the current linear separator, and later on revert to non-zero if badly classified by a next iterate. $t$-ADABOOST makes use of a generalization of classical arithmetic over the reals introduced decades ago [18].

Boosting algorithms for linear models like ADABOOST bring more than just learning good linear separators: it is known that (ada)boosting linear models can be used to emulate the training of *decision trees* (DT) [16], which are models known to lead to some of the best of-the-shelf classifiers when linearly combined [9]. Unsurprisingly, the algorithm obtained emulates the classical top-down induction of a tree found in major packages like CART [6] and C4.5 [23]. The *loss* equivalently minimized, which is, *e.g.*, Matusita's loss for ADABOOST [30, Section 4.1], is a lot more consequential. Contrary to losses for real-valued classification, losses to train DTs rely on the estimates of the posterior learned by the model; they are usually called *losses for Class Probability Estimation* (CPE [25]). The CPE loss is crucial to elicit because (i) it is possible to check whether it is "good" from the standpoint of properness (Bayes rule is optimal for the loss [28]), and (ii) it conditions boosting rates, only a handful of them being known, for the most popular CPE losses [11, 22, 31].

**In this paper**, we show that this emulation scheme on $t$-ADABOOST provides a new family of CPE losses with remarkable constancy with respect to properness: losses are *strictly* proper (Bayes rule is the *sole* optimum) for any $t \in (-\infty, 2)$ and proper for $t = -\infty$. Furthermore, over the range $t \in [-\infty, 1]$, the range of boosting rates spans the full spectrum of known boosting rates [11].

We provide experiments displaying the boosting ability of $t$-ADABOOST over a range of $t$ encompassing potentially more than the set of values covered by our theory, and highlight the potential of using $t$ as a parameter for efficient tuning the loss [25, Section 8]. Due to a lack of space, proofs are relegated to the appendix (APP.). A primer on TEMs is also given in APP., Section I.

## 2   Related work

Boosting refers to the ability of an algorithm to combine the outputs of moderately accurate, "weak" hypotheses into a highly accurate, "strong" ensemble. Originally, boosting was introduced in the context of Valiant's PAC learning model as a way to circumvent the then-existing amount of related negative results [10, 34]. After the first formal proof that boosting is indeed achievable [29], ADABOOST became the first practical and proof-checked boosting algorithm [8, 30]. Boosting was thus born in a machine learning context, but later on, it also emerged in statistics as a way to learn from class residuals computed using the gradient of the loss [9, 21], resulting this time in a flurry of computationally efficient algorithms, still called boosting algorithms, but for which the connection with the original weak/strong learning framework is in general not known.

Our paper draws its boosting connections with ADABOOST's formal lineage. ADABOOST has spurred a long line of work alongside different directions, including statistical consistency [5], noise handling [15, 16], low-resource optimization [22], *etc*. The starting point of our work is a fascinating result in convex optimization establishing a duality between the algorithm and its memory of past iteration's performances, a probability distribution of so-called *weights* over examples [12]. From this standpoint, ADABOOST solves the dual of the optimization of a Bregman divergence (constructed from the negative Shannon entropy as the generator) between weights subject to zero correlation with the last weak classifier's performance. As a consequence, weights define an exponential family. Indeed, whenever a relative entropy is minimized subject to linear constraints, then the solution is a member of an exponential family of distributions (see *e.g.* [2, Section 2.8.1] for an axiomatization of exponential families). ADABOOST's distribution on the examples is a member of a discrete exponential family where the training examples are the finite support of the distribution, sufficient statistics are defined from the weak learners, and the leveraging coefficients are the natural parameters. In summary, there

---

[1]Traditionally, clipping a sum is done after it has been fully computed. In our case, it is clipped after each new summand is added.

is an intimate relationship between boosting à-la-ADABOOST, exponential families, and Bregman divergences [7, 12, 20] and our work "elevates" these methods above exponential families.

# 3 Definitions

We define the $t$-logarithm and $t$-exponential [17, Chapter 7],

$$\log_t(z) \doteq \frac{1}{1-t} \cdot \left(z^{1-t} - 1\right) \quad , \quad \exp_t(z) \doteq [1 + (1-t)z]_+^{1/(1-t)} \quad ([z]_+ \doteq \max\{0, z\}), \quad (1)$$

where the case $t = 1$ is supposed to be the extension by continuity to the $\log$ and $\exp$ functions, respectively. To preserve the concavity of $\log_t$ and the convexity of $\exp_t$, we need $t \geqslant 0$. In the general case, we also note the asymmetry of the composition: while $\exp_t \log_t(z) = z, \forall t \in \mathbb{R}$, we have $\log_t \exp_t(z) = z$ for $t = 1$ ($\forall z \in \mathbb{R}$), but

$$\log_t \exp_t(z) = \max\left\{-\frac{1}{1-t}, z\right\} \quad (t < 1) \quad \text{and} \quad \log_t \exp_t(z) = \min\left\{\frac{1}{t-1}, z\right\} \quad (t > 1).$$

Comparisons between vectors and real-valued functions written on vectors are assumed component-wise. We assume $t \neq 2$ and define notation $t^* \doteq 1/(2-t)$. We now define the key set in which we model our weights (boldfaces denote vector notation).

**Definition 3.1.** *The co-simplex of $\mathbb{R}^m$, $\tilde{\Delta}_m$ is defined as $\tilde{\Delta}_m \doteq \{\boldsymbol{q} \in \mathbb{R}^m : \boldsymbol{q} \geqslant \boldsymbol{0} \wedge \mathbf{1}^\top \boldsymbol{q}^{1/t^*} = 1\}$.*

The letters $\boldsymbol{q}$ will be used to denote TEMs in $\tilde{\Delta}_m$ while $\boldsymbol{p}$ denote the co-density $\boldsymbol{q}^{\frac{1}{t^*}}$ or any element of the probability simplex. We define the general tempered relative entropy as

$$D_t(\boldsymbol{q}'\|\boldsymbol{q}) \quad \doteq \quad \sum_{i \in [m]} q_i' \cdot \left(\log_t q_i' - \log_t q_i\right) - \log_{t-1} q_i' + \log_{t-1} q_i, \quad (2)$$

where $[m] \doteq \{1, ..., m\}$. The tempered relative entropy is a Bregman divergence with convex generator $\varphi_t(z) \doteq z \log_t z - \log_{t-1}(z)$ (for $t \in \mathbb{R}$) and $\varphi_t(z)' = \log_t(x)$. As $t \to 1$, $D_t(\boldsymbol{q}, \boldsymbol{q}')$ becomes the relative entropy with generator $\varphi_1(x) = x \log(x) - x$.

# 4 Tempered boosting as tempered entropy projection

We start with a fixed sample $\mathcal{S} = \{(\boldsymbol{x}_i, y_i) : i \in [m]\}$ where observations $\boldsymbol{x}_i$ lie in some domain $\mathcal{X}$ and labels $y_i$ are $\pm 1$. ADABOOST maintains a distribution $\boldsymbol{p}$ over the sample. At the current iteration, this distribution is updated based on a current *weak hypothesis* $h \in \mathbb{R}^{\mathcal{X}}$ using an exponential update:

$$p_i' = \frac{p_i \cdot \exp(-\mu u_i)}{\sum_k p_k \cdot \exp(-\mu u_k)}, \text{ where } u_i \doteq y_i h(\boldsymbol{x}_i), \mu \in \mathbb{R}.$$

In [12] this update is motivated as minimizing a relative entropy subject to the constraint that $\boldsymbol{p}'$ is a distribution summing to 1 and $\boldsymbol{p}'^\top \boldsymbol{u} = 0$. Following this blueprint, we create a boosting algorithm maintaining a discrete TEM over the sample which is motivated as a constrained minimization of the tempered relative entropy, with a normalization constraint on the co-simplex of $\mathbb{R}^m$:

$$\boldsymbol{q}' \quad \doteq \quad \arg\min_{\substack{\tilde{\boldsymbol{q}} \in \tilde{\Delta}_m \\ \tilde{\boldsymbol{q}}^\top \boldsymbol{u} = 0}} D_t(\tilde{\boldsymbol{q}}\|\boldsymbol{q}), \quad \text{with } \boldsymbol{u} \in \mathbb{R}^m. \quad (3)$$

We now show that the solution $\boldsymbol{q}'$ is a tempered generalization of ADABOOST's exponential update.

**Theorem 1.** *For all $t \in \mathbb{R}\backslash\{2\}$, all solutions to (3) have the form*

$$q_i' = \frac{\exp_t(\log_t q_i - \mu u_i)}{Z_t} \quad \left(= \frac{q_i \otimes_t \exp_t(-\mu u_i)}{Z_t}, \text{ with } a \otimes_t b \doteq [a^{1-t} + b^{1-t} - 1]_+^{\frac{1}{1-t}}\right), \quad (4)$$

*where $Z_t$ ensures co-simplex normalization of the co-density. Furthermore, the unknown $\mu$ satisfies*

$$\mu \in \arg\max -\log_t(Z_t(\mu)) \quad (= \arg\min Z_t(\mu)), \quad (5)$$

**Algorithm 1** $t$-ADABOOST$(t, \mathcal{S}, J)$

---

**Input:** $t \in [0, 1]$, training sample $\mathcal{S}$, #iterations $J$;

**Output:** classifiers $H_J, H_J^{(1/1-t)}$ (see (9));

Step 1 : initialize tempered weights: $\boldsymbol{q}_1 = (1/m^{t*}) \cdot \mathbf{1}$   $(\in \tilde{\Delta}_m)$;

Step 2 : for $j = 1, 2, ..., J$

        Step 2.1 : get weak classifier $h_j \leftarrow$ weak_learner$(\boldsymbol{q}_j, \mathcal{S})$;

        Step 2.2 : choose weight update coefficient $\mu_j \in \mathbb{R}$;

        Step 2.3 : $\forall i \in [m]$, for $u_{ji} \doteq y_i h_j(\boldsymbol{x}_i)$, update the tempered weights as

$$q_{(j+1)i} = \frac{q_{ji} \otimes_t \exp_t(-\mu_j u_{ji})}{Z_{tj}}, \quad \text{where } Z_{tj} = \left\| \boldsymbol{q}_j \otimes_t \exp_t(-\mu_j \boldsymbol{u}_j) \right\|_{1/t*}. \tag{8}$$

        Step 2.4 : choose leveraging coefficient $\alpha_j \in \mathbb{R}$;

---

*or equivalently is a solution to the nonlinear equation*

$$\boldsymbol{q}'(\mu)^\top \boldsymbol{u} = 0. \tag{6}$$

*Finally, if either (i) $t \in \mathbb{R}_{>0} \backslash \{2\}$ or (ii) $t = 0$ and $\boldsymbol{q}$ is not collinear to $\boldsymbol{u}$, then $Z_t(\mu)$ is strictly convex: the solution to (3) is thus unique, and can be found from expression (4) by finding the unique minimizer of (5) or (equivalently) the unique solution to (6).*

(Proof in APP., Section II.1) The $t$-product $\otimes_t$, which satisfies $\exp_t(a + b) = \exp_t(a) \otimes_t \exp_t(b)$, was introduced in [18]. Collinearity never happens in our ML setting because $\boldsymbol{u}$ contains the edges of a weak classifier: $\boldsymbol{q} > 0$ and collinearity would imply that $\pm$ the weak classifier performs perfect classification, and thus defeats the purpose of training an ensemble. $\forall t \in \mathbb{R} \backslash \{2\}$, we have the simplified expression for the normalization coefficient of the TEM and the co-density $\boldsymbol{p}'$ of $\boldsymbol{q}'$:

$$Z_t = \left\| \exp_t (\log_t \boldsymbol{q} - \mu \cdot \boldsymbol{u}) \right\|_{1/t*} \; ; \; p_i' = \frac{p_i \otimes_{t*} \exp_{t*}\left(-\frac{\mu u_i}{t*}\right)}{Z_t'} \; \left( \text{with } Z_t' \doteq Z_t^{1/t*} \right). \tag{7}$$

## 5 Tempered boosting for linear classifiers and clipped linear classifiers

**Models**   A model (or classifier) is an element of $\mathbb{R}^{\mathcal{X}}$. For any model $H$, its empirical risk over $\mathcal{S}$ is $F_{0/1}(H, \mathcal{S}) \doteq (1/m) \cdot \sum_i [\![ y_i \neq \text{sign}(H(\boldsymbol{x}_i)) ]\!]$ where $[\![ . ]\!]$, Iverson's bracket [13], is the Boolean value of the inner predicate. We learn linear separators and *clipped* linear separators. Let $(v_j)_{j \geqslant 1}$ be the terms of a series and $\delta \geqslant 0$. The clipped sum of the series is:

$$\overset{(\delta)}{\underset{(-\delta)}{\sum_{j \in [J]}}} v_j \; \doteq \; \min \left\{ \delta, \max \left\{ -\delta, v_J + \overset{(\delta)}{\underset{(-\delta)}{\sum_{j \in [J-1]}}} v_j \right\} \right\} \quad (\in [-\delta, \delta]), \text{ for } J > 1,$$

and we define the base case $J = 1$ by replacing the inner clipped sum with 0. Note that clipped summation is non-commutative, and so is different from clipping in $[-\delta, \delta]$ the whole sum itself[2]. Given a set of so-called weak hypotheses $h_j \in \mathbb{R}^{\mathcal{X}}$ and leveraging coefficients $\alpha_j \in \mathbb{R}$ (for $j \in [J]$), the corresponding linear separators and clipped linear separators are

$$H_J(\boldsymbol{x}) \doteq \sum_{j \in [J]} \alpha_j h_j(\boldsymbol{x}) \quad ; \quad H_J^{(\delta)}(\boldsymbol{x}) \doteq \overset{(\delta)}{\underset{(-\delta)}{\sum_{j \in [J]}}} \alpha_j h_j(\boldsymbol{x}). \tag{9}$$

**Tempered boosting and its general convergence**   Our algorithm, $t$-ADABOOST, is presented in Algorithm 1, using presentation conventions from [30]. Before analyzing its convergence, several properties are to be noted for $t$-ADABOOST: first, it keeps the appealing property, introduced by ADABOOST, that examples receiving the wrong class by the current weak classifier are reweighted

---

[2]Fix for example $a = -1, b = 3, \delta = 2$. For $v_1 = a, v_2 = b$, the clipped sum is $2 = -1 + 3$, but for $v_1 = b, v_2 = a$, the clipped sum becomes $1 = \mathbf{2} - 1$.

higher (if $\mu_j > 0$). Second, the leveraging coefficients for weak classifiers in the final classifier ($\alpha_j$s) are not the same as the ones used to update the weights ($\mu_j$s), unless $t = 1$. Third and last, because of the definition of $\exp_t$ (1), if $t < 1$, tempered weights can switch off and on, *i.e.*, become 0 if an example is "too well classified" and then revert back to being $> 0$ if the example becomes wrongly classified by the current weak classifier (if $\mu_j > 0$). To take into account those zeroing weights, we denote $[m]_j^\dagger \doteq \{i : q_{ji} = 0\}$ and $m_j^\dagger \doteq \mathrm{Card}([m]_j^\dagger)$ ($\forall j \in [J]$ and $\mathrm{Card}$ denotes the cardinal). Let $R_j \doteq \max_{i \notin [m]_j^\dagger} |y_i h_j(\boldsymbol{x}_i)|/q_{ji}^{1-t}$ and $q_j^\dagger \doteq \max_{i \in [m]_j^\dagger} |y_i h_j(\boldsymbol{x}_i)|^{1/(1-t)}/R_j^{1/(1-t)}$. It is worth noting that $q_j^\dagger$ is homogeneous to a tempered weight.

**Theorem 2.** *At iteration $j$, define the weight function $q'_{ji} \doteq q_{ji}$ if $i \notin [m]_j^\dagger$ and $q_j^\dagger$ otherwise; set*

$$\rho_j \quad \doteq \quad \frac{1}{(1 + m_j^\dagger q_j^{\dagger 2-t}) R_j} \cdot \sum_{i \in [m]} q'_{ji} y_i h_j(\boldsymbol{x}_i) \quad (\in [-1, 1]). \tag{10}$$

*In algorithm $t$-ADABOOST, consider the choices (with the convention $\prod_{k=1}^0 v_k \doteq 1$)*

$$\mu_j \doteq -\frac{1}{R_j} \cdot \log_t \left( \frac{1 - \rho_j}{M_{1-t}(1 - \rho_j, 1 + \rho_j)} \right) \quad , \quad \alpha_j \doteq m^{1-t*} \cdot \left( \prod_{k=1}^{j-1} Z_k \right)^{1-t} \cdot \mu_j, \tag{11}$$

*where $M_q(a, b) \doteq ((a^q + b^q)/2)^{1/q}$ is the $q$-power mean. Then for any $H \in \{H_J, H_J^{(1/1-t)}\}$, its empirical risk is upperbounded as:*

$$F_{0/1}(H, \mathcal{S}) \leqslant \prod_{j=1}^J Z_{tj}^{2-t} \leqslant \prod_{j=1}^J \left( 1 + m_j^\dagger q_j^{\dagger 2-t} \right) \cdot K_t(\rho_j) \quad \left( K_t(z) \doteq \frac{1 - z^2}{M_{1-t}(1 - z, 1 + z)} \right). \tag{12}$$

(Proof in APP., Section II.2) We jointly comment $t$-ADABOOST and Theorem 2 in two parts.

**Case $t \to 1^-$:** $t$-ADABOOST converges to ADABOOST and Theorem 2 to its convergence analysis: $t$-ADABOOST converges to ADABOOST as presented in [30, Figure 1]: the tempered simplex becomes the probability simplex, $\otimes_t$ converges to regular multiplication, weight update (8) becomes ADABOOST's, $\alpha_j \to \mu_j$ in (11) and finally the expression of $\mu_j$ converges to ADABOOST's leveraging coefficient in [30] ($\lim_{t \to 1} M_{1-t}(a, b) = \sqrt{ab}$). Even guarantee (12) converges to ADABOOST's popular guarantee of [30, Corollary 1] ($\lim_{t \to 1} K_t(z) = \sqrt{1 - z^2}$, $m_j^\dagger = 0$). Also, in this case, we learn only the unclipped classifier since $\lim_{t \to 1^-} H_J^{(1/1-t)} = H_J$.

**Case $t < 1$:** Let us first comment on the convergence rate. The proof of Theorem 2 shows that $K_t(z) \leqslant \exp(-z^2/(2t*))$. Suppose there is no weight switching, so $m_j^\dagger = 0, \forall j$ (see Section 7) and, as in the boosting model, suppose there exists $\gamma > 0$ such that $|\rho_j| \geqslant \gamma, \forall j$. Then $t$-ADABOOST is guaranteed to attain empirical risk below some $\varepsilon > 0$ after a number of iterations equal to $J = (2t*/\gamma^2) \cdot \log(1/\varepsilon)$. $t*$ being an increasing function of $t \in [0, 1]$, we see that $t$-ADABOOST is able to slightly improve upon ADABOOST's celebrated rate [32]. However, $t* = 1/2$ for $t = 0$ so the improvement is just on the hidden constant. This analysis is suited for small values of $|\rho_j|$ and does not reveal an interesting phenomenon for better weak hypotheses. Figure 1 compares $K_t(z)$ curves ($K_1(z) \doteq \lim_{t \to 1} K_t(z) = \sqrt{1 - z^2}$ for ADABOOST, see [30, Corollary 1]), showing the case $t < 1$ can be substantially better, especially when weak hypotheses are not "too weak". If $m_j^\dagger > 0$, switching weights can impede our convergence *analysis*, though exponential convergence is always possible if $m_j^\dagger q_j^{\dagger 2-t}$ is small enough; also, when it is not, we may in fact have converged to a good model (see APP., Remark 1). A good criterion to train weak hypotheses is then the optimization of the edge $\rho_j$, thus using $\boldsymbol{q}'_j$ normalized in the simplex. Other key features of $t$-ADABOOST are as follows. First, the weight update and leveraging coefficients of weak classifiers are bounded because $|\mu_j| < 1/(R_j(1 - t))$ (APP., Lemma H) (this is not the case for $t \to 1^-$). This guarantees that new weights are bounded before normalization (unlike for $t \to 1^-$). Second, we remark that $\mu_j \neq \alpha_j$ if $t \neq 1$. Factor $m^{1-t*}$ is added for convergence analysis purposes; we can discard it to train the unclipped classifier: it does not change its empirical risk. This is, however, different for factor $\prod_{k=1}^{j-1} Z_k$: from (12), we conclude that this is an indication of how well the past ensemble performs.

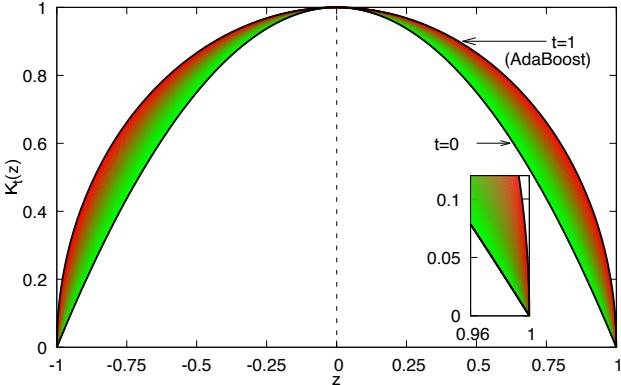

Figure 1: Plot of $K_t(z)$ in (12), $t \in [0,1]$ (the smaller, the better for convergence).

As it gets better and better, it progressively dampens the leverage of the next weak classifiers, a phenomenon that does not occur in boosting, where an excellent weak hypothesis on the current weights can have a leveraging coefficient so large that it wipes out the classification of the past ones. This can be useful to control numerical instabilities.

**Extension to margins**   A key property of boosting algorithms like ADABOOST is to be able to boost not just the empirical risk but more generally *margins* [19, 30], where a margin integrates both the accuracy of label prediction but also the confidence in prediction (say $|H|$). We generalize the margin notion of [19] to the tempered arithmetic and let $\nu_t((\boldsymbol{x}, y), H) \doteq \tanh_t(yH(\boldsymbol{x})/2)$ denote the margin of $H$ on example $(\boldsymbol{x}, y)$, where $\tanh_t(z) \doteq (1 - \exp_t(-2z))/(1 + \exp_t(-2z))(\in [-1, 1])$ is the tempered hyperbolic tangent. The objective of minimizing the empirical risk is generalized to minimizing the margin risk, $F_{t,\theta}(H, \mathcal{S}) \doteq (1/m) \cdot \sum_i [\![\nu_t((\boldsymbol{x}_i, y_i), H) \leqslant \theta]\!]$, where $\theta \in (-1, 1)$. Guarantees on the empirical risk are guarantees on the margin risk for $\theta = 0$ only. In just a few steps, we can generalize Theorem 2 to *all* $\theta \in (-1, 1)$. For space reason, we state the core part of the generalization, from which extending it to a generalization of Theorem 2 is simple.

**Theorem 3.** *For any $\theta \in (-1, 1)$ and $t \in [0, 1]$, the guarantee of algorithm $t$-*ADABOOST *in Theorem 2 extends to the margin risk, with notations from Theorem 2, via:*

$$F_{t,\theta}(H, \mathcal{S}) \quad \leqslant \quad \left( \frac{1 + \theta}{1 - \theta} \right)^{2-t} \prod_{j=1}^{J} Z_{tj}^{2-t}. \tag{13}$$

(Proof in APP., Section II.3) At a high level, $t$-ADABOOST brings similar margin maximization properties as ADABOOST. Digging a bit in (13) reveals an interesting phenomenon for $t \neq 1$ on how margins are optimized compared to $t = 1$. Pick $\theta < 0$, so we focus on those examples for which the classifier $H$ has high confidence in its *wrong* classification. In this case, factor $((1 + \theta)/(1 - \theta))^{2-t}$ is increasing as a function of $t \in [0, 1]$ (and this pattern is reversed for $\theta > 0$). In words, the smaller we pick $t \in [0, 1]$ and the better is the bound in (13), suggesting increased "focus" of $t$-ADABOOST on increasing the margins of examples *with low negative margin* (*e.g.* the most difficult ones) compared to the case $t = 1$.

**The tempered exponential loss**   In the same way as ADABOOST introduced the now famous exponential loss, (12) recommends to minimize the normalization coefficient, following (7),

$$Z_{tj}^{2-t}(\mu) \quad = \quad \left\| \exp_t \left( \log_t \boldsymbol{q}_j - \mu \cdot \boldsymbol{u}_j \right) \right\|_{1/t*}^{1/t*} \quad (\text{with } u_{ji} \doteq y_i h_j(\boldsymbol{x}_i)). \tag{14}$$

We cannot easily unravel the normalization coefficient to make appear an equivalent generalization of the exponential loss, unless we make several assumptions, one being $\max_i |h_j(\boldsymbol{x}_i)|$ is small enough for any $j \in [J]$. In this case, we end up with an equivalent criterion to minimize which looks like

$$F_t(H, \mathcal{S}) \quad = \quad \frac{1}{m} \cdot \sum_i \exp_t^{2-t} (-y_i H(\boldsymbol{x}_i)), \tag{15}$$

where we have absorbed in $H$ the factor $m^{1-t^*}$ appearing in the $\exp_t$ (scaling $H$ by a positive value does not change its empirical risk). This defines a generalization of the exponential loss which we call the *tempered exponential loss*. Notice that one can choose to minimize $F_t(H, \mathcal{S})$ disregarding any constraint on $|H|$.

## 6 A broad family of boosting-compliant proper losses for decision trees

**Losses for class probability estimation**  When it comes to tabular data, it has long been known that some of the best models to linearly combine with boosting are decision trees (DT, [9]). Decision trees, like other domain-partitioning classifiers, are not trained by minimizing a *surrogate loss* defined over real-valued predictions, but defined over *class probability estimation* (CPE, [26]), those estimators being posterior estimation computed at the leaves. Let us introduce a few definitions for those. A CPE loss $\ell : \{-1, 1\} \times [0, 1] \to \mathbb{R}$ is

$$\ell(y, u) \quad \dot{=} \quad [\![ y = 1 ]\!] \cdot \ell_1(u) + [\![ y = -1 ]\!] \cdot \ell_{-1}(u). \tag{16}$$

Functions $\ell_1, \ell_{-1}$ are called *partial* losses. The pointwise conditional risk of local guess $u \in [0, 1]$ with respect to a ground truth $v \in [0, 1]$ is:

$$L(u, v) \quad \dot{=} \quad v \cdot \ell_1(u) + (1 - v) \cdot \ell_{-1}(u). \tag{17}$$

A loss is *proper* iff for any ground truth $v \in [0, 1]$, $L(v, v) = \inf_u L(u, v)$, and strictly proper iff $u = v$ is the sole minimizer [26]. The (pointwise) *Bayes* risk is $\underline{L}(v) \doteq \inf_u L(u, v)$. The log/cross-entropy-loss, square-loss, Matusita loss are examples of CPE losses. One then trains a DT minimizing the expectation of this loss over leaves' posteriors, $\mathbb{E}_\lambda[\underline{L}(p_\lambda)]$, $p_\lambda$ being the local proportion of positive examples at leaf $\lambda$ – or equivalently, the local posterior.

**Deriving CPE losses from (ada)boosting**  Recently, it was shown how to derive in a general way a CPE loss to train a DT from the minimization of a surrogate loss with a boosting algorithm [16]. In our case, the surrogate would be $\tilde{Z}_{tj}$ (14) and the boosting algorithm, $t$-ADABOOST. The principle is simple and fits in four steps: (i) show that a DT can equivalently perform simple linear classifications, (ii) use a weak learner that designs splits and the boosting algorithm to fit the leveraging coefficient and compute those in closed form, (iii) simplify the expression of the loss using those, (iv) show that the expression simplified is, in fact, a CPE loss. To get (i), we remark that a DT contains a tree (graph). One can associate to each node a real value. To classify an observation, we sum all reals from the root to a leaf and decide on the class based on the sign of the prediction, just like for any real-valued predictor. Suppose we are at a leaf. What kind of weak hypotheses can create splits "in disguise"? Those can be of the form

$$h_j(\boldsymbol{x}) \quad \dot{=} \quad [\![ x_k \geqslant a_j ]\!] \cdot b_j, \quad a_j, b_j \in \mathbb{R},$$

where the observation variable $x_k$ is assumed real valued for simplicity and the test $[\![ x_k \geqslant a_j ]\!]$ splits the leaf's domain in two non-empty subsets. This creates half of the split. $\overline{h}_j(\boldsymbol{x}) \doteq [\![ x_k < a_j ]\!] \cdot -b_j$ creates the other half of the split. Remarkably $h_j$ satisfies the weak learning assumption iff $\overline{h}_j$ does [16]. So we get the split design part of (ii). We compute the leveraging coefficients at the new leaves from the surrogate's minimization / boosting algorithm, end up with new real predictions at the new leaves (instead of the original $b_j, -b_j$), push those predictions in the surrogate loss for (iii), simplify it and, quite remarkably end up with a loss of the form $\mathbb{E}_\lambda[\mathrm{L}(p_\lambda)]$, where L turns out to be the pointwise Bayes risk $\underline{L}$ of a proper loss [16] (notation from [26]).

In the case of [16], it is, in fact, granted that we end up with such a "nice" CPE loss because of the choice of the surrogates at the start. In our case, however, nothing grants this *a priori* if we start from the tempered exponential loss $\tilde{Z}_{tj}$ (14) so it is legitimate to wonder whether such a chain of derivations (summarized) can happen to reverse engineer an interesting CPE loss:

$$\tilde{Z}_{tj} \overset{?}{\mapsto} \mathrm{L} \overset{?}{\mapsto} \underline{L}^{(t)} \overset{?}{\mapsto} \ell_1^{(t)}; \ell_{-1}^{(t)} \quad \text{(proper ? strictly proper ? for which } ts \text{ ?, ...)} \tag{18}$$

When such a complete derivation happens until the partial losses $\ell_1; \ell_{-1}$ and their properties, we shall write that minimizing $\tilde{Z}_{tj}$ *elicits* the corresponding loss and partial losses.

**Theorem 4.** *Minimizing $\tilde{Z}_{tj}$ elicits the CPE loss we define as the **tempered loss**, with partial losses:*

$$\ell_1^{(t)}(u) \doteq \left( \frac{1 - u}{M_{1-t}(u, 1 - u)} \right)^{2 - t} \quad , \quad \ell_{-1}^{(t)}(u) \doteq \ell_1^{(t)}(1 - u), \quad (t \in [-\infty, 2]). \tag{19}$$

*The tempered loss is symmetric, differentiable, strictly proper for $t \in (-\infty, 2)$ and proper for $t = -\infty$.*

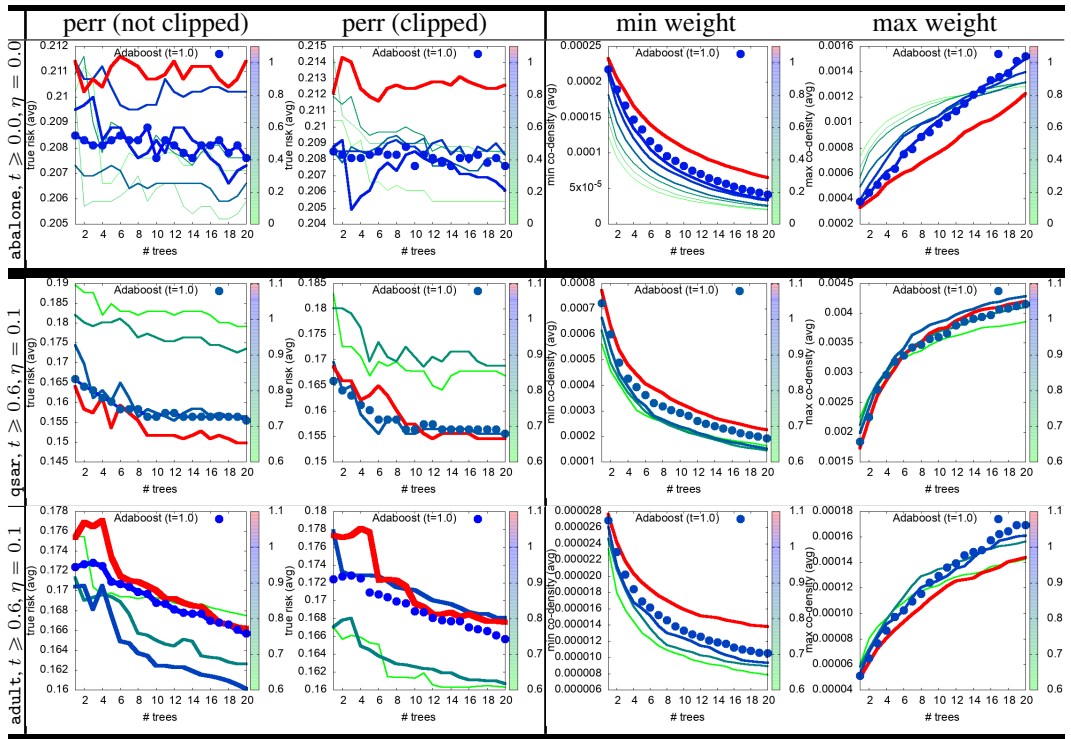

Table 1: Experiments on $t$-ADABOOST comparing with ADABOOST ($t = 1$, bullets) on three domains (rows), displaying from left to right the estimated true error of non-clipped and clipped models, and the min and max codensity weights. These domains were chosen to give an example of three different situations: small values of $t$ perform well (`abalone`), the best performance is achieved by the largest $t$ (*e.g.* ADABOOST, `qsar`), and the worst performance is achieved by the largest $t$ (`adult`). Topmost row is without noise ($\eta = 0$) while the others are with $10\%$ training noise; $t$ scale displayed with varying color and width (colormap indicated on each plot). Averages shown for readability: see Table 2 for exhaustive statistical tests.

Differentiability means the partial losses are differentiable, and symmetry follows from the relationship between partial losses [20] (the proof, in APP., Section II.4, derives the infinite case, $\ell_1^{(-\infty)}(u) = 2 \cdot [\![u \leqslant 1/2]\!]$) Let us explicit the Bayes risk of the tempered loss and a key property.

**Lemma 1.** *The Bayes risk of the tempered loss is ($M_q$ defined in Theorem 2):*

$$\underline{L}^{(t)}(u) = \frac{2u(1-u)}{M_{1-t}(u, 1-u)}, \tag{20}$$

*and it satisfies* $\forall u \in [0,1], \forall z \in [2 \cdot \min\{u, 1-u\}, 1], \exists t \in [-\infty, 2]$ *such that* $\underline{L}^{(t)}(u) = z$.

Lemma 1, whose proof is trivial, allows to show a key boosting result: $t = 1$ retrieves Matusita's loss, for which a near-optimal boosting rate is known [11] while $t = -\infty$ retrieves the empirical risk, which yields the worst possible guarantee [11]. In between, we have, for example, CART's Gini criterion for $t = 0$, which yields an intermediate boosting guarantee. Continuity with respect to $t$ of the Bayes risks in between the empirical risk and Matusita's loss means the boosting ranges of the tempered loss cover *the full known spectrum of boosting rates* for $t \in [-\infty, 1]$. We know of no (differentiable and) proper CPE loss with such coverage. Note that (i) this is a non-constructive result as we do not associate a specific $t$ for a specific rate, and (ii) the state-of-the-art boosting rates for DT induction do not seem to cover the case $t \in (1, 2)$, thus left as an open question.

## 7 Experiments

We have performed experiments on a testbed of 10 UCI domains, whose details are given in APP. (Section A3). Experiments were carried out using a 10-fold stratified cross-validation procedure.

To compare $t$-ADABOOST with ADABOOST, we ran $t$-ADABOOST with a first range of values of $t \in \{0.0, 0.2, 0.4, 0.6, 0.8, 0.9\}$. This is in the range of values covered by our convergence result for linear separators in Theorem 2. Our results on decision tree induction cover a much wider range, in particular for $t \in (1, 2)$. To assess whether this can be an interesting range to study, we added $t = 1.1$ to the set of tested $t$ values. When $t > 1$, some extra care is to be put into computations because the weight update becomes unbounded, in a way that is worse than ADABOOST. Indeed, as can be seen from (8), if $\mu_j y_i h_j(\boldsymbol{x}_i) \leqslant -1/(t-1)$ (the example is badly classified by the current weak hypothesis, assuming wlog $\mu_j > 0$), the weight becomes infinity before renormalization. In our experiments, picking a value of $t$ close to 2 clearly shows this problem, so to be able to still explore whether $t > 1$ can be useful, we picked a value close to 1, namely $t = 1.1$, and checked that in our experiments this produced no such numerical issue. We also considered training clipped and not clipped models.

All boosting models were trained for a number of $J = 20$ decision trees (The appendix provides experiments on training bigger sets). Each decision tree is induced using the tempered loss with the corresponding value of $t$ (see Theorem 4) following the classical top-down template, which consists in growing the current heaviest leaf in the tree and picking the best split for the leaf chosen. We implemented $t$-ADABOOST exactly as in Section 5, including computing leveraging coefficients as suggested. Thus, we do not scale models. More details are provided in the appendix. In our experiments, we also included experiments on a phenomenon highlighted more than a decade ago [15] and fine-tuned more recently [16], the fact that a convex booster's model is the weakest link when it has to deal with noise in training data. This is an important task because while the tempered exponential loss is convex, it does not fit into the blueprint loss of [15, Definition 1] because it is not $C^1$ if $t \neq 1$. One might thus wonder how $t$-ADABOOST behaves when training data is affected by noise. Letting $\eta$ denote the proportion of noisy data in the training sample, we tried $\eta \in \{0.0, 0.1\}$ (The appendix provides experiments on more noise levels). We follow the noise model of [15] and thus independently flip the true label of each example with probability $\eta$.

For each run, we recorded the average test error and the average maximum and minimum co-density weight. Table 1 presents a subset of the results obtained on three domains. Table 2 presents a more synthetic view in terms of statistical significance of the results for $t \neq 1$ vs. $t = 1$ (ADABOOST). The table reports only results for $t \geqslant 0.6$ for synthesis purposes. Values $t < 0.6$ performed on average slightly worse than the others *but* on some domains, as the example of `abalone` suggests in Table 2 (the plots include all values of $t$ tested in $[0, 1.1]$), we clearly got above-par results for such small values of $t$, both in terms of final test error but also fast early convergence to low test error. This comment can be generalized to all values of $t$.

The weights reveal interesting patterns as well. First, perhaps surprisingly, we never encountered the case where weights switch off, regardless of the value of $t$. The average minimum weight curves of Table 1 generalize to all our tests (see the appendix). This does not rule out the fact that boosting for a much longer number of iterations might lead to weights switching off/on, but the fact that this does not happen at least early during boosting probably comes from the fact that the leveraging coefficients for weights ($\mu_.$) are bounded. Furthermore, their maximal absolute value is all the smaller as $t$ decreases to 0. Second, there is a pattern that also repeats on the maximum weights, not on all domains but on a large majority of them and can be seen in `abalone` and `adult` in Table 1: the maximum weight of ADABOOST tends to increase much more rapidly compared to $t$-ADABOOST with $t < 1$. In the latter case, we almost systematically observe that the maximum weight tends to be upperbounded, which is not the case for ADABOOST (the growth of the maximal weight looks almost linear). Having bounded weights could be of help to handle numerical issues of (ada)boosting [14].

Our experiments certainly confirm the boosting nature of $t$-ADABOOST if we compare its convergence to that of ADABOOST: more often than not, it is in fact comparable to that of ADABOOST. While this applies broadly for $t \geqslant 0.6$, we observed examples where much smaller values (even $t = 0.0$) could yield such fast convergence. Importantly, this applies to clipped models as well and it is important to notice because it means attaining a low "boosted" error does not come at the price of learning models with large range. This is an interesting property: for $t = 0.0$, we would be guaranteed that the computation of the clipped prediction is always in $[-1, 1]$. Generalizing our comment on small values of $t$ above, we observed that an efficient tuning algorithm for $t$ could be able to get very substantial leverage over ADABOOST. Table 2 was crafted for a standard limit $p$-val of 0.1 and "blurs" the best results that can be obtained. On several domains (`winered`, `abalone`, `eeg`, `creditcard`, `adult`), applicable $p$-values for which we would conclude that some $t \neq 1$ performs better than $t = 1$ drop in

| $\eta$ | 0.0 | | | | | | | | 0.1 | | | | | | | |
|---|---|---|---|---|---|---|---|---|---|---|---|---|---|---|---|---|
| $t$ | 0.6 | | 0.8 | | 0.9 | | 1.1 | | 0.6 | | 0.8 | | 0.9 | | 1.1 | |
| $[\![\text{clipped}]\!]$ | 0 | 1 | 0 | 1 | 0 | 1 | 0 | 1 | 0 | 1 | 0 | 1 | 0 | 1 | 0 | 1 |
| #better | 2 | 3 | 1 | 2 | 1 | 3 | | | 1 | 1 | 1 | 2 | 2 | 1 | | |
| #equivalent | 5 | 5 | 6 | 6 | 7 | 7 | 6 | 7 | 4 | 8 | 8 | 7 | 8 | 9 | 8 | 10 |
| #worse | 3 | 2 | 3 | 2 | 2 | | 4 | 3 | 5 | 1 | 1 | 1 | | | 2 | |

Table 2: Outcomes of student paired $t$-tests over 10 UCI domains, with training noise $\eta \in \{0.0, 0.1\}$, for $t \in \{0.6, 0.8, 0.9, 1.0, 1.1\}$ and with / without clipped models. For each triple ($\eta$, $t$, $[\![\text{clipped}]\!]$), we give the number of domains for which the corresponding setting of $t$-ADABOOST is statistically better than ADABOOST(#better), the number for which it is statistically worse (#worse) and the number for which we cannot reject the assumption of identical performances. Threshold $p-\text{val} = 0.1$.

between $7E − 4$ and $0.05$. Unsurprisingly, ADABOOST also manages to beat significantly alternative values of $t$ in several cases. Our experiments with training noise ($\eta = 0.1$) go in the same direction. Looking at Table 1, one could eventually be tempted to conclude that $t$ slightly smaller than 1.0 may be a better choice than adaboosting ($t = 1$), as suggested by our results for $t = 0.9$, but we do not think this produces a general "rule-of-thumb". There is also no apparent "noise-dependent" pattern that would obviously separate the cases $t < 1$ from $t = 1$ even when the tempered exponential loss does not fit to [15]'s theory. Finally, looking at the results for $t > 1$ also yields the same basic conclusions, which suggests that boosting can be attainable outside the range covered by our theory (in particular Theorem 2).

All this brings us to the experimental conclusion that the question does not reside on opposing the case $t \neq 1$ to the case $t = 1$. Rather, our experiments suggest – pretty much like our theory does – that the actual question resides in how to efficiently *learn* $t$ on a domain-dependent basis. Our experiments indeed demonstrate that substantial gains could be obtained, to handle overfitting or noise.

## 8 Discussion, limitations and conclusion

ADABOOST is one of the original and simplest Boosting algorithms. In this paper we generalized ADABOOST to maintaining a tempered measure over the examples by minimizing a tempered relative entropy. We kept the setup as simple as possible and therefore focused on generalizing ADABOOST. However more advanced boosting algorithms have been designed based on relative entropy minimization subject to linear constraints. There are versions that constrain the edges of all past hypotheses to be zero [36]. Also, when the maximum margin of the game is larger than zero, then ADABOOST cycles over non-optimal solutions [27]. Later Boosting algorithms provably optimize the margin of the solution by adjusting the constraint value on the dual edge away from zero (see e.g. [24]). Finally, the ELRP-Boost algorithm optimizes a trade off between relative entropy and the edge [35]. We conjecture that all of these orthogonal direction have generalizations to the tempered case as well and are worth exploring.

These are theoretical directions that, if successful, would contribute to bring more tools to the design of rigorous boosting algorithms. This is important because boosting suffers several impediments, not all of which we have mentioned: for example, to get statistical consistency for ADABOOST, it is known that early stopping is mandatory [5]. More generally, non-Lipschitz losses like the exponential loss seem to be harder to handle compared to Lipschitz losses [33] (but they yield in general better convergence rates). The validity of the weak learning assumption of boosting can also be discussed, in particular regarding the negative result of [15] which advocates, beyond just better (ada)boosting, for boosting for *more* classes of models / architectures [16]. Alongside this direction, we feel that our experiments on noise handling give a preliminary account of the fact that there is no "one $t$ fits all" case, but a much more in depth analysis is required to elicit / tune a "good" $t$. This is a crucial issue for noise handling [16], but as we explain in Section 7, this could bring benefits in much wider contexts as well.

## Acknowledgments

The authors thank the reviewers for numerous comments that helped improving the paper's content.

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

# Appendix

## Abstract

This is the Appendix to Paper "Boosting with Tempered Exponential Measures". To differentiate with the numberings in the main file, the numbering of Theorems, Lemmata, Definitions is letter-based (A, B, ...).

## Table of contents

# I  A short primer on Tempered Exponential Measures

We describe here the minimal amount of material necessary to understand how our approach to boosting connects to these measures. We refer to [4] for more details. With a slight abuse of notation, we define the perspective transforms $(\log_t)^*(z) \doteq t^* \cdot \log_{t*}(z/t^*)$ and $(\exp_t)^*(z) \doteq t^* \cdot \exp_{t*}(z/t^*)$. Recall that $t^* \doteq 1/(2-t)$.

**Definition A.** *[4] A tempered exponential measure (TEM) family is a set of unnormalized densities in which each element admits the following canonical expression:*

$$q_{t|\boldsymbol{\theta}}(\boldsymbol{x}) \doteq \frac{\exp_t(\boldsymbol{\theta}^\top \boldsymbol{\varphi}(\boldsymbol{x}))}{\exp_t(G_t(\boldsymbol{\theta}))} = \exp_t(\boldsymbol{\theta}^\top \boldsymbol{\varphi}(\boldsymbol{x}) \ominus_t G_t(\boldsymbol{\theta})) \quad \left( a \ominus_t b \doteq \frac{a-b}{1+(1-t)b} \right), \qquad (21)$$

*where $\boldsymbol{\theta}$ is the element's natural parameter, $\boldsymbol{\varphi}(\boldsymbol{x})$ is the sufficient statistics and*

$$G_t(\boldsymbol{\theta}) = (\log_t)^* \int (\exp_t)^*(\boldsymbol{\theta}^\top \boldsymbol{\varphi}(\boldsymbol{x})) \mathrm{d}\xi$$

*is the (convex) cumulant, $\xi$ being a base measure (implicit).*

Except for $t = 1$ (which reduces a TEM family to a classical exponential family), the total mass of a TEM is not 1 (but it has an elegant closed form expression [4]). However, the exponentiated $q_{t|\boldsymbol{\theta}}^{1/t^*}$ does sum to 1. In the discrete case, this justifies extending the classical simplex to what we denote as the co-simplex.

**Definition B.** *The co-simplex of $\mathbb{R}^m$, $\tilde{\Delta}_m$ is defined as $\tilde{\Delta}_m \doteq \{\boldsymbol{q} \in \mathbb{R}^m : \boldsymbol{q} \geqslant \boldsymbol{0} \wedge \boldsymbol{1}^\top \boldsymbol{q}^{1/t^*} = 1\}$.*

The connection between $t$-ADABOOST's update and TEM's is immediate from the equation's update ((4) in MF). We can show that $\tilde{\Delta}_m$ can also be represented as TEMs.

**Lemma A.** *$\tilde{\Delta}_m$ is a (discrete) family of tempered exponential measures.*

*Proof.* We proceed as in [2, Section 2.2.2] for exponential families: let $\boldsymbol{q} \in \tilde{\Delta}_m$, which we write

$$q(n) \doteq \sum_{i \in [m]} q_i \cdot [\![ i = n ]\!], n \in [m]. \qquad (22)$$

$[\![ \pi ]\!]$, the Iverson bracket [13], takes value 1 if Boolean predicate $\pi$ is true (and 0 otherwise). We create $m - 1$ natural parameters and the cumulant,

$$\theta_i \doteq \log_t \frac{q_i}{q_m}, i \in [m-1] \quad ; \quad G_t(\boldsymbol{\theta}) \doteq \log_t \frac{1}{q_m},$$

and end up with (22) also matching the atom mass function

$$q(n) = \frac{\exp_t \left( \sum_{i \in [m-1]} \theta_i \cdot [\![ i = n ]\!] \right)}{\exp_t G_t(\boldsymbol{\theta})},$$

which clearly defines a tempered exponential measure over $[m]$. This ends the proof of Lemma A. $\qquad \square$

## II   Supplementary material on proofs

### II.1   Proof of Theorem 1

To improve readability, we remove dependency in $t$ in normalization coefficient $Z$. We use notations from [4, proof of Theorem 3.2] and denote the Lagrangian

$$\mathcal{L} \;=\; \Delta(\tilde{\boldsymbol{q}}\|\boldsymbol{q}) + \lambda \left(\sum_i \tilde{q}_i^{1/t^*} - 1\right) - \sum_i \nu_i \tilde{q}_i + \mu \sum_i \tilde{q}_i u_i, \tag{23}$$

which yields $\partial \mathcal{L}/\partial \tilde{q}_i = \log_t \tilde{q}_i - \log_t q_i + \lambda \tilde{q}_i^{1-t} - \nu_i + \mu u_i$ ($\lambda$ absorbs factor $2-t$), and, rearranging (absorbing factor $1-t$ in $\nu_i$),

$$(1 + (1-t)\lambda)\tilde{q}_i^{1-t} \;=\; \nu_i + 1 + (1-t)(\log_t q_i - \mu u_i), \forall i \in [m]. \tag{24}$$

We see that $\lambda \neq -1/(1-t)$ otherwise the Lagrangian drops its dependence in the unknown. In fact, the solution necessarily has $1 + (1-t)\lambda > 0$. To see this, we distinguish two cases: (i) if some $u_k = 0$, then since $\log_t q_k \geqslant -1/(1-t)$ there would be no solution to (24) if $1 + (1-t)\lambda < 0$ because of the KKT conditions $\nu_i \geqslant 0, \forall i \in [m]$; (ii) otherwise, if all $u_k \neq 0, \forall k \in [m]$, then there must be two coordinates of different signs otherwise there is no solution to our problem (3) (main file, we must have indeed $\tilde{\boldsymbol{q}} \geqslant 0$ because of the co-simplex constraint). Thus, there exists at least one coordinate $k \in [m]$ for which $-(1-t)\mu u_k > 0$ and since $\log_t q_k \geqslant -1/(1-t)$ (definition of $\log_t$) and $\nu_k \geqslant 0$ (KKT conditions), the RHS of (24) for $i = k$ is $> 0$, preventing $1 + (1-t)\lambda < 0$ in the LHS.

We thus have $1 + (1-t)\lambda > 0$. The KKT conditions ($\nu_i \geqslant 0, \nu_i \tilde{q}_i = 0, \forall i \in [m]$) yield the following: $1 + (1-t)(\log_t q_i - \mu u_i) > 0$ imply $\nu_i = 0$ and $1 + (1-t)(\log_t q_i - \mu u_i) \leqslant 0$ imply $\tilde{q}_i^{1-t} = 0$ so we get the necessary form for the optimum:

$$\begin{aligned}
\tilde{q}_i &= \frac{\exp_t(\log_t q_i - \mu u_i)}{\exp_t \lambda} \\
&= \frac{q_i \otimes_t \exp_t(-\mu u_i)}{Z_t}, \tag{25}
\end{aligned}$$

where $\lambda$ or $Z_t \doteq \exp_t \lambda$ ensures normalisation for the co-density. Note that we have a simplified expression for the co-density:

$$p_i \;=\; \frac{p_{ji} \otimes_{t*} \exp_{t*}(-\mu u_i/t^*)}{Z_t^{\mathrm{co}}}, \tag{26}$$

with $Z_t^{\mathrm{co}} \doteq Z_t^{1/t^*} = \sum_i p_{ji} \otimes_{t*} \exp_{t*}(-\mu u_i/t^*)$. For the analytic form in (25), we can simplify the Lagrangian to a dual form that depends on $\mu$ solely:

$$\mathcal{D}(\mu) \;=\; \Delta(\tilde{\boldsymbol{q}}(\mu)\|\boldsymbol{q}) + \mu \sum_i \tilde{q}_i(\mu) u_i. \tag{27}$$

The proof of (5) (main file) is based on a key Lemma.

**Lemma B.** *For any $\tilde{\boldsymbol{q}}$ having form (25) such that $\tilde{\boldsymbol{q}}^\top \boldsymbol{u} = 0$, $\mathcal{D}(\mu) = -\log_t Z_t(\mu)$.*

*Proof.* For any $\tilde{\boldsymbol{q}}$ having form (25), denote

$$[m]_* \;\doteq\; \{i : \tilde{q}_i \neq 0\}. \tag{28}$$

We first compute (still using $\lambda \doteq \log_t Z_t(\mu)$ for short):

$$
\begin{aligned}
A &\doteq \sum_i \tilde{q}_i \cdot \log_t \tilde{q}_i \\
&= \sum_{i \in [m]_*} \tilde{q}_i \cdot \log_t \left( \frac{\exp_t (\log_t q_i - \mu u_i)}{\exp_t \lambda} \right) \\
&= \sum_{i \in [m]_*} \tilde{q}_i \cdot \left( \frac{1}{1-t} \cdot \left[ \frac{1 + (1-t)(\log_t q_i - \mu u_i)}{1 + (1-t)\lambda} - 1 \right] \right) \\
&= \frac{1}{1-t} \cdot \sum_{i \in [m]_*} \tilde{q}_i \cdot \left( \frac{q_i^{1-t} - (1-t)\mu u_i}{1 + (1-t)\lambda} \right) - \frac{1}{1-t} \cdot \sum_{i \in [m]_*} \tilde{q}_i \\
&= -\frac{\mu}{1 + (1-t)\lambda} \cdot \sum_{i \in [m]_*} \tilde{q}_i u_i + \frac{1}{(1-t)(1 + (1-t)\lambda)} \cdot \sum_{i \in [m]_*} \tilde{q}_i q_i^{1-t} - \frac{1}{1-t} \cdot \sum_{i \in [m]_*} \tilde{q}_i \\
&= \underbrace{-\frac{\mu}{1 + (1-t)\lambda} \cdot \tilde{q}^\top u}_{\doteq B} + \underbrace{\frac{1}{(1-t)(1 + (1-t)\lambda)} \cdot \sum_{i \in [m]} \tilde{q}_i q_i^{1-t}}_{\doteq C} - \underbrace{\frac{1}{1-t} \cdot \sum_{i \in [m]} \tilde{q}_i}_{\doteq D}. \quad (29)
\end{aligned}
$$

Remark that in the last identity, we have put back summations over the complete set $[m]$ of indices. We note that $B = 0$ because $\tilde{q}^\top u = 0$. We then remark that without replacing the expression of $\tilde{q}$, we have in general for any $\tilde{q} \in \tilde{\Delta}_m$:

$$
\begin{aligned}
E &\doteq \sum_{i \in [m]} \tilde{q}_i \cdot (\log_t \tilde{q}_i - \log_t q_i) \\
&= \sum_{i \in [m]} \tilde{q}_i \cdot \left( \frac{1}{1-t} \cdot (\tilde{q}_i^{1-t} - 1) - \frac{1}{1-t} \cdot (q_i^{1-t} - 1) \right) \\
&= \frac{1}{1-t} \cdot \sum_{i \in [m]} \tilde{q}_i^{2-t} - \frac{1}{1-t} \cdot \sum_{i \in [m]} \tilde{q}_i q_i^{1-t} \\
&= \frac{1}{1-t} \cdot \left( 1 - \sum_{i \in [m]} \tilde{q}_i q_i^{1-t} \right),
\end{aligned}
$$

and we can check that for any $\tilde{q}, q \in \tilde{\Delta}_m$, $E = \Delta(\tilde{q}\|q)$. We then develop $\Delta(\tilde{q}\|q)$ with a partial replacement of $\tilde{q}$ by its expression:

$$
\begin{aligned}
\Delta(\tilde{q}\|q) &= A - \sum_i \tilde{q}_i \log_t q_i \\
&= A - \frac{1}{1-t} \cdot \sum_i \tilde{q}_i q_i^{1-t} + \frac{1}{1-t} \cdot \sum_i \tilde{q}_i \\
&= C - \frac{1}{1-t} \cdot \sum_i \tilde{q}_i q_i^{1-t} \\
&= \frac{1}{1-t} \cdot \left( \frac{1}{1 + (1-t)\lambda} - 1 \right) \cdot \sum_i \tilde{q}_i q_i^{1-t} \\
&= -\frac{\lambda}{1 + (1-t)\lambda} \cdot \sum_i \tilde{q}_i q_i^{1-t} \\
&= -\frac{\lambda}{1 + (1-t)\lambda} \cdot (1 - (1-t) \cdot \Delta(\tilde{q}\|q)).
\end{aligned}
$$

Rearranging gives that for any $\tilde{q}, q \in \tilde{\Delta}_m$ such that (i) $\tilde{q}$ has the form (25) for some $\mu \in \mathbb{R}$ and (ii) $\tilde{q}^\top u = 0$,

$$
\begin{aligned}
\Delta(\tilde{q}\|q) &= -\lambda \\
&= -\log_t(Z_t),
\end{aligned}
$$

as claimed. This ends the proof of Lemma B. $\qquad\square$

We thus get from the definition of the dual that $\mu = \arg\max - \log_t Z_t(\mu) = \arg\min Z_t(\mu)$. We have the explicit form for $Z_t$:

$$
\begin{aligned}
Z_t(\mu) &= \left( \sum_i \exp_t^{2-t} \left( \log_t q_i - \mu u_i \right) \right)^{\frac{1}{2-t}} \\
&= \left( \sum_{i \in [m]_*} \exp_t^{2-t} \left( \log_t q_i - \mu u_i \right) \right)^{\frac{1}{2-t}},
\end{aligned}
$$

where $[m]_*$ is defined in (28). We remark that the last expression is differentiable in $\mu$, and get

$$
\begin{aligned}
Z_t'(\mu) &= \frac{1}{2-t} \cdot \left( \sum_{i \in [m]_*} \exp_t^{2-t} \left( \log_t q_i - \mu u_i \right) \right)^{-\frac{1-t}{2-t}} \\
&\quad \cdot (2-t) \sum_{i \in [m]_*} \exp_t^{1-t} \left( \log_t q_i - \mu u_i \right) \cdot \exp_t^t \left( \log_t q_i - \mu u_i \right) \cdot -u_i \\
&= -Z_t^{t-1} \cdot \sum_{i \in [m]_*} \exp_t \left( \log_t q_i - \mu u_i \right) \cdot u_i \\
&= -Z_t^t \cdot \sum_{i \in [m]_*} \tilde{q}_i u_i \\
&= -Z_t^t \cdot \sum_{i \in [m]} \tilde{q}_i u_i \\
&= -Z_t^t \cdot \tilde{q}^\top u, \qquad (30)
\end{aligned}
$$

so

$$
\begin{aligned}
\frac{\partial - \log_t (Z_t)}{\partial \mu} &= -Z_t^{-t} Z_t' \\
&= \tilde{q}(\mu)^\top u,
\end{aligned}
$$

and we get that any critical point of $Z_t(\mu)$ satisfies $\tilde{q}(\mu)^\top u = 0$. A sufficient condition to have just one critical point, being the minimum sought is the strict convexity of $Z_t(\mu)$. The next Lemma provides the proof that it is for all $t > 0$.

**Lemma C.** $Z_t''(\mu) \geqslant t \cdot Z_t(\mu)^{2t-1} (\tilde{q}(\mu)^\top u)^2$.

*Proof.* After simplifications, we have

$$
\begin{aligned}
Z_t^{3-2t} \cdot Z_t'' &= (t-1) \cdot \left( \sum_{i \in [m]} \exp_t \left( \log_t q_i - \mu u_i \right) \cdot u_i \right)^2 \qquad (31) \\
&\quad + \left( \sum_{i \in [m]} \exp_t^{2-t} \left( \log_t q_i - \mu u_i \right) \right) \cdot \left( \sum_{i \in [m]} \exp_t^t \left( \log_t q_i - \mu u_i \right) \cdot u_i^2 \right) \quad (32) \\
&= (t-1) \cdot \sum_{i,k \in [m]} Q_i Q_k u_i u_k + \sum_{i,k \in [m]} Q_i^{2-t} Q_k^t u_k^2, \qquad (33)
\end{aligned}
$$

where we have let $Q_i \doteq \exp_t \left( \log_t q_i - \mu u_i \right) \geqslant 0$. Since $a^2 + b^2 \geqslant 2ab$, we note that for any $i \neq k$,

$$
\begin{aligned}
Q_i^{2-t} Q_k^t u_k^2 + Q_k^{2-t} Q_i^t u_i^2 &\geqslant 2\sqrt{Q_i^{2-t} Q_k^t Q_k^{2-t} Q_i^t} u_i u_k \\
&= 2 Q_i Q_k u_i u_k, \qquad (34)
\end{aligned}
$$

so we split (33) in two terms and get

$$
\begin{aligned}
Z_t^{3-2t} \cdot Z_t'' &= (t-1) \cdot \sum_{i \in [m]} Q_i^2 u_i^2 + \sum_{i \in [m]} Q_i^{2-t} Q_i^t u_i^2 \\
&\quad + \sum_{i,k \in [m], i<k} 2(t-1) Q_i Q_k u_i u_k + \sum_{i,k \in [m], i<k} Q_i^{2-t} Q_k^t u_k^2 + Q_k^{2-t} Q_i^t u_i^2 \\
&= t \cdot \sum_{i \in [m]} Q_i^2 u_i^2 \\
&\quad + \sum_{i,k \in [m], i<k} 2(t-1) Q_i Q_k u_i u_k + \sum_{i,k \in [m], i<k} Q_i^{2-t} Q_k^t u_k^2 + Q_k^{2-t} Q_i^t u_i^2 \\
&\geqslant t \cdot \sum_{i \in [m]} Q_i^2 u_i^2 + 2t \cdot \sum_{i,k \in [m], i<k} Q_i Q_k u_i u_k \quad\quad (35) \\
&= t \cdot \left( \sum_{i \in [m]} \exp_t \left( \log_t q_i - \mu u_i \right) \cdot u_i \right)^2 \\
&= t Z_t^2 \cdot (\tilde{\boldsymbol{q}}^\top \boldsymbol{u})^2, \quad\quad (36)
\end{aligned}
$$

where we have used (34) in (35). Since $Z_t(\mu) > 0$, we get the statement of Lemma C after reorganising (36). □

Lemma C shows the strict convexity of $Z_t(\mu)$ for any $t > 0$. The case $t = 0$ follows by direct differentiation: we get after simplification

$$
Z_t''(\mu) = \frac{\left( \sum_{i \in [m]} u_i^2 \right) \cdot \left( \sum_{i \in [m]} (q_i - \mu u_i)^2 \right) - \left( \sum_{i \in [m]} (q_i - \mu u_i) u_i \right)^2}{\left( \sum_{i \in [m]} (q_i - \mu u_i)^2 \right)^{\frac{3}{2}}}.
$$

Cauchy-Schwartz inequality allows to conclude that $Z_t''(\mu) \geqslant 0$ and is in fact $> 0$ *unless* $\tilde{\boldsymbol{q}}$ is collinear to $\boldsymbol{u}$. This completes the proof of Theorem 1.

## II.2  Proof of Theorem 2

The proof involves several arguments, organized into several subsections. Some are more general than what is strictly needed for the proof of the Theorem, on purpose.

### II.II.2.1  Clipped summations

For any $\delta \geqslant 0$, we define clipped summations of the sequence of ordered elements $v_1, v_2, ..., v_J$: if $J > 1$,

$$
{}^{(\delta)}\sum_{j=1}^J v_j \doteq \min \left\{ v_J + {}^{(\delta)}\sum_{j=1}^{J-1} v_j, \delta \right\} \quad , \quad {}_{(-\delta)}\sum_{j=1}^J v_j \doteq \max \left\{ v_J + {}_{(-\delta)}\sum_{j=1}^{J-1} v_j, -\delta \right\}, \quad (37)
$$

and the base case ($J = 1$) is obtained by replacing the inner sum by 0. We also define the doubly clipped summation:

$$
{}^{(\delta)}_{(-\delta)}\sum_{j=1}^J v_j \doteq \max \left\{ \min \left\{ v_J + {}^{(\delta)}_{(-\delta)}\sum_{j=1}^{J-1} v_j, \delta \right\}, -\delta \right\},
$$

with the same convention for the base case. We prove a series of simple but useful properties of the clipped summation.

**Lemma D.** *The following properties hold true for clipped summation:*

1. *(doubly) clipped summations are noncommutative;*

2. *(doubly) clipped summations are ordinary summation in the limit: for any $J \geqslant 1$ and any sequence $v_1, v_2, ..., v_J$,*

$$\lim_{\delta \to +\infty} {}^{(\delta)}\sum_{j=1}^{J} v_j = \lim_{\delta \to +\infty} {}_{(-\delta)}\sum_{j=1}^{J} v_j = \lim_{\delta \to +\infty} {}^{(\delta)}_{(-\delta)}\sum_{j=1}^{J} v_j = \sum_{j=1}^{J} v_j$$

3. *clipped summations sandwich ordinary summation and the doubly clipped summation: for any $\delta \geqslant 0$, any $J \geqslant 1$ and any sequence $v_1, v_2, ..., v_J$,*

$${}^{(\delta)}\sum_{j=1}^{J} v_j \leqslant \sum_{j=1}^{J} v_j \leqslant {}_{(-\delta)}\sum_{j=1}^{J} v_j \quad ; \quad {}^{(\delta)}\sum_{j=1}^{J} v_j \leqslant {}^{(\delta)}_{(-\delta)}\sum_{j=1}^{J} v_j \leqslant {}_{(-\delta)}\sum_{j=1}^{J} v_j$$

*Proof.* Noncommutativity follows from simple counterexamples: for example, for $v \doteq -1$ and $w \doteq 2$, if we fix $v_1 \doteq v, v_2 \doteq w$, then ${}^{(0)}\sum_{j=1}^{2} v_j = 1$ while ${}^{(0)}\sum_{j=1}^{2} v_{3-j} = -1$. Property [2.] is trivial. The set of leftmost inequalities of property [3.] can be shown by induction, noting the base case is trivial and otherwise, using the induction hypothesis in the leftmost inequality,

$${}^{(\delta)}\sum_{j=1}^{J+1} v_j \doteq \min\left\{ v_{J+1} + {}^{(\delta)}\sum_{j=1}^{J} v_j, \delta \right\} \leqslant \min\left\{ v_{J+1} + \sum_{j=1}^{J} v_j, \delta \right\} \leqslant v_{J+1} + \sum_{j=1}^{J} v_j = \sum_{j=1}^{J+1} v_j,$$

and similarly

$$\begin{aligned}
{}_{(-\delta)}\sum_{j=1}^{J+1} v_j &\doteq \max\left\{ v_{J+1} + {}_{(-\delta)}\sum_{j=1}^{J} v_j, -\delta \right\} \\
&\geqslant \max\left\{ v_{J+1} + \sum_{j=1}^{J} v_j, -\delta \right\} \geqslant v_{J+1} + \sum_{j=1}^{J} v_j = \sum_{j=1}^{J+1} v_j.
\end{aligned}$$

A similar argument holds for the set of rightmost inequalities: for example, the induction's general case holds

$$\begin{aligned}
{}^{(\delta)}\sum_{j=1}^{J+1} v_j &\doteq \min\left\{ v_{J+1} + {}^{(\delta)}\sum_{j=1}^{J} v_j, \delta \right\} \\
&\leqslant \min\left\{ v_{J+1} + {}^{(\delta)}_{(-\delta)}\sum_{j=1}^{J} v_j, \delta \right\} \\
&\leqslant \max\left\{ \min\left\{ v_{J+1} + {}^{(\delta)}_{(-\delta)}\sum_{j=1}^{J} v_j, \delta \right\}, -\delta \right\} = {}^{(\delta)}_{(-\delta)}\sum_{j=1}^{J} v_j.
\end{aligned}$$

for the leftmost inequality. This ends the proof of Lemma D. $\square$

### II.II.2.2 Unravelling weights

**Lemma E.** *Define*

$$v_j \doteq m^{1-t^*} \cdot \left( \prod_{k=1}^{j-1} Z_{tk} \right)^{1-t} \cdot \mu_j \quad \left( \text{convention} : \prod_{k=1}^{0} u_k \doteq 1 \right). \tag{38}$$

*Then $\forall J \geqslant 1$, weights unravel as:*

$$q_{(J+1)i} = \begin{cases} \dfrac{1}{m^{t^*} \prod_{j=1}^{J} Z_{tj}} \cdot \exp_t\left( - {}^{(1/1-t)}\sum_{j=1}^{J} v_j u_{ji} \right) & \text{if} \quad t < 1 \\[2em] \dfrac{1}{m^{t^*} \prod_{j=1}^{J} Z_{tj}} \cdot \exp_t\left( - {}_{(-1/1-t)}\sum_{j=1}^{J} v_j u_{ji} \right) & \text{if} \quad t > 1 \end{cases}.$$

*Proof.* We start for the case $t < 1$. We proceed by induction, noting first that the normalization constraint for the initial weights imposes $q_{1i} = 1/m^{1/(2-t)} = 1/m^{t*}$ and so (using $(1-t)t* = 1-t*$)

$$
\begin{aligned}
q_{2i} &= \frac{\exp_t(\log_t q_{1i} - \mu_1 u_{1i})}{Z_1} \\[1mm]
&= \frac{1}{Z_1} \cdot \left[ 1 + (1-t) \cdot \left( \frac{1}{1-t} \cdot \left( \frac{1}{m^{\frac{1-t}{2-t}}} - 1 \right) - \mu_1 u_{1i} \right) \right]_+^{\frac{1}{1-t}} \\[1mm]
&= \frac{1}{Z_1} \cdot \left[ \frac{1}{m^{1-t*}} - (1-t)\mu_1 u_{1i} \right]_+^{\frac{1}{1-t}} \\[1mm]
&= \frac{1}{m^{t*} Z_1} \cdot \left[ 1 - (1-t)m^{1-t*}\mu_1 u_{1i} \right]_+^{\frac{1}{1-t}} \\[1mm]
&= \frac{1}{m^{t*} Z_1} \cdot \exp_t \left( -^{(1/1-t)} \sum_{j=1}^{1} v_j u_{ji} \right),
\end{aligned}
$$

completing the base case. Using the induction hypothesis, we unravel at iteration $J + 1$:

$$
\begin{aligned}
&q_{(J+1)i} \\[1mm]
&= \frac{\exp_t(\log_t q_{Ji} - \mu_J u_{Ji})}{Z_J} \\[1mm]
&= \frac{\exp_t \left( \log_t \left( \frac{1}{m^{t*} \prod_{j=1}^{J-1} Z_{tj}} \cdot \exp_t \left( -^{(1/1-t)} \sum_{j=1}^{J-1} v_j u_{ji} \right) \right) - \mu_J u_{Ji} \right)}{Z_J} \\[1mm]
&= \frac{\exp_t \left( \log_t \exp_t \left( -^{(1/1-t)} \sum_{j=1}^{J-1} v_j u_{ji} \right) \ominus_t \log_t \left( m^{t*} \prod_{j=1}^{J-1} Z_{tj} \right) - \mu_J u_{Ji} \right)}{Z_J} \\[1mm]
&= \frac{1}{Z_J} \cdot \exp_t \left( \frac{\max\left\{ -\frac{1}{1-t}, -^{(1/1-t)} \sum_{j=1}^{J-1} v_j u_{ji} \right\} - \log_t \left( m^{t*} \prod_{j=1}^{J-1} Z_{tj} \right)}{1 + (1-t)\log_t \left( m^{t*} \prod_{j=1}^{J-1} Z_{tj} \right)} - \mu_J u_{Ji} \right) \\[1mm]
&= \frac{1}{Z_J} \cdot \left[ 1 + \frac{(1-t)\cdot\max\left\{ -\frac{1}{1-t}, -^{(1/1-t)} \sum_{j=1}^{J-1} v_j u_{ji} \right\} - (1-t)\log_t \left( m^{t*} \prod_{j=1}^{J-1} Z_{tj} \right)}{1 + (1-t)\log_t \left( m^{t*} \prod_{j=1}^{J-1} Z_{tj} \right)} - (1-t)\mu_J u_{Ji} \right]_+^{\frac{1}{1-t}} \\[1mm]
&= \frac{1}{Z_J} \cdot \left[ 1 + \frac{(1-t)\cdot\max\left\{ -\frac{1}{1-t}, -^{(1/1-t)} \sum_{j=1}^{J-1} v_j u_{ji} \right\} - \left( \left( m^{t*} \prod_{j=1}^{J-1} Z_{tj} \right)^{1-t} - 1 \right)}{\left( m^{t*} \prod_{j=1}^{J-1} Z_{tj} \right)^{1-t}} - (1-t)\mu_J u_{Ji} \right]_+^{\frac{1}{1-t}},
\end{aligned}
$$

which simplifies into (using $(1-t)t* = 1 - t*$)

$$
\begin{aligned}
&q_{(J+1)i} \\[1mm]
&= \frac{1}{m^{t*} \prod_{j=1}^{J} Z_{tj}} \cdot \left[ 1 + (1-t) \cdot \left( \max\left\{ -\frac{1}{1-t}, -^{(1/1-t)} \sum_{j=1}^{J-1} v_j u_{ji} \right\} - v_J u_{Ji} \right) \right]_+^{\frac{1}{1-t}} \quad (39) \\[1mm]
&= \frac{1}{m^{t*} \prod_{j=1}^{J} Z_{tj}} \cdot \exp_t (-S_J),
\end{aligned}
$$

with

$$S_J \doteq \min\left\{-\max\left\{-\frac{1}{1-t}, -\,{}^{(1/1-t)}\!\!\sum_{j=1}^{J-1} v_j u_{ji}\right\} + v_J u_{Ji}, \frac{1}{1-t}\right\}$$

$$= \min\left\{v_J u_{Ji} + \min\left\{\frac{1}{1-t}, {}^{(1/1-t)}\!\!\sum_{j=1}^{J-1} v_j u_{ji}\right\}, \frac{1}{1-t}\right\}$$

$$= \min\left\{v_J u_{Ji} + {}^{(1/1-t)}\!\!\sum_{j=1}^{J-1} v_j u_{ji}, \frac{1}{1-t}\right\}$$

$$\doteq {}^{(1/1-t)}\!\!\sum_{j=1}^{J} v_j u_{ji}$$

(we used twice the definition of clipped summation), which completes the proof of Lemma E for $t < 1$.

We now treat the case $t > 1$. The base induction is equivalent, while unraveling gives, instead of (39):

$$q_{(J+1)i}$$

$$= \frac{1}{m^{t*}\prod_{j=1}^{J} Z_{tj}} \cdot \left[1 + (1-t)\cdot\left(\min\left\{-\frac{1}{1-t}, -\,{}^{-(1/t-1)}\!\!\sum_{j=1}^{J-1} v_j u_{ji}\right\} - v_J u_{Ji}\right)\right]_+^{\frac{1}{1-t}}$$

$$= \frac{1}{m^{t*}\prod_{j=1}^{J} Z_{tj}} \cdot \exp_t(-S_J),$$

and, this time,

$$S_J \doteq \max\left\{-\min\left\{-\frac{1}{1-t}, -\,{}^{-(1/t-1)}\!\!\sum_{j=1}^{J-1} v_j u_{ji}\right\} + v_J u_{Ji}, -\frac{1}{t-1}\right\} \tag{40}$$

$$= \max\left\{v_J u_{Ji} + \max\left\{-\frac{1}{t-1}, {}^{-(1/t-1)}\!\!\sum_{j=1}^{J-1} v_j u_{ji}\right\}, -\frac{1}{t-1}\right\} \tag{41}$$

$$= \max\left\{v_J u_{Ji} + {}^{-(1/t-1)}\!\!\sum_{j=1}^{J-1} v_j u_{ji}, -\frac{1}{t-1}\right\} \tag{42}$$

$$\doteq {}_{(-1/t-1)}\!\!\sum_{j=1}^{J} v_j u_{ji}, \tag{43}$$

which completes the proof of Lemma E. $\qquad\square$

### II.II.2.3 Introducing classifiers

**Ordinary linear separators**   Suppose we have a classifier

$$H_J(\boldsymbol{x}) \doteq \sum_{j=1}^{J} \beta_j^{1-t} \mu_j \cdot h_j(\boldsymbol{x}), \quad \beta_j \doteq m^{t*}\prod_{k=1}^{j-1} Z_{tk},$$

where $\mu_j \in \mathbb{R}, \forall j \in [J]$. We remark that $[\![z \neq r]\!] \leqslant \exp_t^{2-t}(-zr)$ for any $t \leqslant 2$ and $z, r \in \mathbb{R}$, and $z \mapsto \exp_t^{2-t}(-z)$ is decreasing for any $t \leqslant 2$, so using [3.] in Lemma D, we get for our training

sample $\mathcal{S} \doteq \{(\boldsymbol{x}_i, y_i), i \in [m]\}$ and any $t < 1$ (from Lemma E),

$$\frac{1}{m} \cdot \sum_{i \in [m]} [\![\operatorname{sign}(H_J(\boldsymbol{x}_i)) \neq y_i]\!]$$

$$\leqslant \sum_{i \in [m]} \frac{\exp_t^{2-t}\left(-\sum_{j=1}^{J} m^{1-t*}\left(\prod_{k=1}^{j-1} Z_{tk}\right)^{1-t} \mu_j \cdot y_i h_j(\boldsymbol{x}_i)\right)}{m}$$

$$\leqslant \sum_{i \in [m]} \frac{\exp_t^{2-t}\left(-^{(1/1-t)}\sum_{j=1}^{J} m^{1-t*}\left(\prod_{k=1}^{j-1} Z_{tk}\right)^{1-t} \mu_j \cdot y_i h_j(\boldsymbol{x}_i)\right)}{m} \quad (44)$$

$$= \sum_{i \in [m]} \frac{\exp_t^{2-t}\left(-^{(1/1-t)}\sum_{j=1}^{J} v_j u_{ji}\right)}{m},$$

where

$$v_j \doteq m^{1-t*}\left(\prod_{k=1}^{j-1} Z_{tk}\right)^{1-t} \mu_j \quad ; \quad u_{ji} \doteq y_i h_j(\boldsymbol{x}_i). \quad (45)$$

Using Lemma E with those definitions, we get

$$\frac{1}{m} \cdot \sum_{i \in [m]} [\![\operatorname{sign}(H_J(\boldsymbol{x}_i)) \neq y_i]\!] \leqslant \sum_{i \in [m]} \frac{\left(q_{(J+1)i} m^{t*} \prod_{j=1}^{J} Z_{tj}\right)^{2-t}}{m}$$

$$= \prod_{j=1}^{J} Z_{tj}^{2-t} \cdot \sum_{i \in [m]} q_{(J+1)i}^{2-t}$$

$$= \prod_{j=1}^{J} Z_{tj}^{2-t}, \quad (46)$$

because $q_J \in \tilde{\Delta}_m$. We thus have proven the following Lemma.

**Lemma F.** *For any $t < 1$ and any linear separator*

$$H_J(\boldsymbol{x}) \quad \doteq \quad \sum_{j=1}^{J} \beta_j^{1-t} \mu_j \cdot h_j(\boldsymbol{x}), \quad \left(\beta_j \doteq m^{t*} \prod_{k=1}^{j-1} Z_{tk}, \mu_j \in \mathbb{R}, h_j \in \mathbb{R}^{\mathcal{X}}, \forall j \in [J]\right),$$

*where $Z_{tk}$ is the normalization coefficient of $\boldsymbol{q}$ in (25) with $u_{ji} \doteq y_i h_j(\boldsymbol{x}_i)$,*

$$\frac{1}{m} \cdot \sum_{i \in [m]} [\![\operatorname{sign}(H_J(\boldsymbol{x}_i)) \neq y_i]\!] \leqslant \prod_{j=1}^{J} Z_{tj}^{2-t}. \quad (47)$$

**Clipped linear separators** Suppose we have a classifier ($t < 1$)

$$H_J^{(1/1-t)}(\boldsymbol{x}) \quad \doteq \quad ^{(1/1-t)}_{(-1/1-t)}\sum_{j=1}^{J} \beta_j^{1-t} \mu_j \cdot h_j(\boldsymbol{x}), \quad \beta_j \doteq m^{t*} \prod_{k=1}^{j-1} Z_{tk}.$$

We can now replace (44) by

$$\frac{1}{m} \cdot \sum_{i\in[m]} [\![\text{sign}(H_J^{(1/1-t)}(\boldsymbol{x}_i)) \neq y_i]\!]$$

$$\leq \sum_{i\in[m]} \frac{\exp_t^{2-t}\left(-y_i \cdot {}^{(1/1-t)}_{(-1/1-t)}\!\sum_{j=1}^{J} m^{1-t*}\left(\prod_{k=1}^{j-1} Z_{tk}\right)^{1-t}\mu_j \cdot h_j(\boldsymbol{x}_i)\right)}{m}$$

$$= \sum_{i\in[m]} \frac{\exp_t^{2-t}\left(-{}^{(1/1-t)}_{(-1/1-t)}\!\sum_{j=1}^{J} m^{1-t*}\left(\prod_{k=1}^{j-1} Z_{tk}\right)^{1-t}\mu_j \cdot y_i h_j(\boldsymbol{x}_i)\right)}{m}$$

$$\leq \sum_{i\in[m]} \frac{\exp_t^{2-t}\left(-{}^{(1/1-t)}\!\sum_{j=1}^{J} m^{1-t*}\left(\prod_{k=1}^{j-1} Z_{tk}\right)^{1-t}\mu_j \cdot y_i h_j(\boldsymbol{x}_i)\right)}{m} \tag{48}$$

$$= \sum_{i\in[m]} \frac{\exp_t^{2-t}\left(-{}^{(1/1-t)}\!\sum_{j=1}^{J} v_j u_{ji}\right)}{m}.$$

The first identity has used the fact that $y_i \in \{-1,1\}$, so it can be folded in the doubly clipped summation without changing its value, and the second inequality used [3.] in Lemma D. This directly leads us to the following Lemma.

**Lemma G.** *For any $t < 1$ and any clipped linear separator*

$$H_J^{(1/1-t)}(\boldsymbol{x}) \;\doteq\; {}^{(1/1-t)}_{(-1/1-t)}\!\sum_{j=1}^{J} \beta_j^{1-t}\mu_j \cdot h_j(\boldsymbol{x}), \quad \left(\beta_j \doteq m^{t*}\prod_{k=1}^{j-1} Z_{tk}, \mu_j \in \mathbb{R}, h_j \in \mathbb{R}^{\mathcal{X}}, \forall j \in [J]\right),$$

*where $Z_{tk}$ is the normalization coefficient of $\boldsymbol{q}$ in (25) with $u_{ji} \doteq y_i h_j(\boldsymbol{x}_i)$,*

$$\frac{1}{m} \cdot \sum_{i\in[m]} [\![\text{sign}(H_J^{(1/1-t)}(\boldsymbol{x}_i)) \neq y_i]\!] \;\leq\; \prod_{j=1}^{J} Z_{tj}^{2-t}. \tag{49}$$

### II.II.2.4 Geometric convergence of the empirical risk

To get the right-hand side of (47) and (49) as small as possible, we can independently compute each $\mu_j$ so as to minimize

$$Z_{tj}^{2-t}(\mu) \;\doteq\; \sum_{i\in[m]} \exp_t^{2-t}\left(\log_t q_{ji} - \mu u_{ji}\right). \tag{50}$$

We proceed in two steps, first computing a convenient upperbound for (50), and then finding the $\mu$ that minimizes this upperbound.

**Step 1**: We distinguish two cases depending on weight $q_{ji}$. Let $[m]_j^+ \doteq \{i : q_{ji} > 0\}$ and $[m]_j^\dagger \doteq \{i : q_{ji} = 0\}$:

**Case 1** $i \in [m]_j^+$. Let $r_{ji} \doteq u_{ji}/q_{ji}^{1-t}$ and suppose $R_j > 0$ is a real that satisfies

$$|r_{ji}| \;\leq\; R_j, \forall i \in [m]_j^+. \tag{51}$$

For any convex function $f$ defined on $[-1,1]$, we have $f(z) \leq ((1+z)/2) \cdot f(1) + ((1-z)/2) \cdot f(-1), \forall z \in [-1,1]$ (the straight line is the chord crossing $f$ at $z = -1, 1$). Because

$z \mapsto [1-z]_+^{\frac{2-t}{1-t}}$ is convex for $t \leqslant 2$, for any $i \in [m]_j^+$

$$\exp_t^{2-t}\left(\log_t q_{ji} - \mu u_{ji}\right)$$

$$= \left[q_{ji}^{1-t} - (1-t)\mu u_{ji}\right]_+^{\frac{2-t}{1-t}}$$

$$= q_{ji}^{2-t} \cdot \left[1 - (1-t)\mu R_j \cdot \frac{r_{ji}}{R_j}\right]_+^{\frac{2-t}{1-t}}$$

$$\leqslant q_{ji}^{2-t} \cdot \frac{R_j + r_{ji}}{2R_j}\left[1 - (1-t)\mu R_j\right]_+^{\frac{2-t}{1-t}} + q_{ji}^{2-t} \cdot \frac{R_j - r_{ji}}{2R_j}\left[1 + (1-t)\mu R_j\right]_+^{\frac{2-t}{1-t}}$$

$$= \frac{q_{ji}^{2-t}R_j + q_{ji}u_{ji}}{2R_j}\left[1 - (1-t)\mu R_j\right]_+^{\frac{2-t}{1-t}} + \frac{q_{ji}^{2-t}R_j - q_{ji}u_{ji}}{2R_j}\left[1 + (1-t)\mu R_j\right]_+^{\frac{2-t}{1-t}}.$$

**Case 2** $i \in [m]_j^\dagger$. Let $q_j^\dagger > 0$ be a real that satisfies

$$\frac{|u_{ji}|}{q_j^{\dagger 1-t}} < R_j, \forall i \in [m]_j^\dagger. \tag{52}$$

Using the same technique as in case 1, we find for any $i \in [m]_j^\dagger$

$$\exp_t^{2-t}\left(\log_t q_{ji} - \mu u_{ji}\right)$$

$$= \exp_t^{2-t}\left(-\frac{1}{1-t} - \mu u_{ji}\right)$$

$$= \left[-(1-t)\mu u_{ji}\right]_+^{\frac{2-t}{1-t}}$$

$$\leqslant \left[q_j^{\dagger 1-t} - (1-t)\mu u_{ji}\right]_+^{\frac{2-t}{1-t}}$$

$$\leqslant \frac{q_j^{\dagger 2-t}R_j + q_j^\dagger u_{ji}}{2R_j}\left[1 - (1-t)\mu R_j\right]_+^{\frac{2-t}{1-t}} + \frac{q_j^{\dagger 2-t}R_j - q_j^\dagger u_{ji}}{2R_j}\left[1 + (1-t)\mu R_j\right]_+^{\frac{2-t}{1-t}}.$$

Folding both cases into one and letting

$$q'_{ji} \doteq \begin{cases} q_{ji} & \text{if} \quad i \in [m]_j^+ \\ q_j^\dagger & \text{if} \quad i \in [m]_j^\dagger \end{cases}, \tag{53}$$

we get after summation, using $m_j^\dagger \doteq \mathrm{Card}([m]_j^\dagger)$ and

$$\rho_j \doteq \frac{1}{(1 + m_j^\dagger q_j^{\dagger 2-t})R_j} \cdot \sum_{i \in [m]} q'_{ji}u_{ji} \quad (\in [-1,1]), \tag{54}$$

that

$$Z_{tj}^{2-t}(\mu)$$

$$\leqslant \frac{(1 + m_j^\dagger q_j^{\dagger 2-t})R_j}{2R_j} \cdot \left((1+\rho_j)\left[1 - (1-t)\mu R_j\right]_+^{\frac{2-t}{1-t}} + (1-\rho_j)\left[1 + (1-t)\mu R_j\right]_+^{\frac{2-t}{1-t}}\right)$$

$$= \frac{1 + m_j^\dagger q_j^{\dagger 2-t}}{2} \cdot \left((1+\rho_j)\cdot\exp_t^{2-t}(-\mu R_j) + (1-\rho_j)\cdot\exp_t^{2-t}(\mu R_j)\right). \tag{55}$$

**Step 2**: we have our upperbound for (50). We now compute the minimizer $\mu^*$ of (55). If this minimizer satisfies

$$|\mu^*| < \frac{1}{R_j|1-t|}, \tag{56}$$

then it can be found by ordinary differentiation, as the solution to

$$(1-\rho_j)\cdot\exp_t(\mu^* R_j) - (1+\rho_j)\cdot\exp_t(-\mu^* R_j) = 0,$$

which is equivalently

$$\frac{\exp_t\left(-\mu^* R_j\right)}{\exp_t\left(\mu^* R_j\right)} = \exp_t\left(-\mu^* R_j \ominus_t \mu^* R_j\right)$$

$$= \frac{1-\rho_j}{1+\rho_j},$$

where we recall $a \ominus_t b \doteq (a-b)/(1+(1-t)b)$. Solving it yields

$$\mu^* = \frac{1}{R_j} \cdot -\frac{1}{1-t} \cdot \left(\frac{(1-\rho_j)^{1-t} - (1+\rho_j)^{1-t}}{(1-\rho_j)^{1-t} + (1+\rho_j)^{1-t}}\right)$$

$$= \frac{1}{R_j} \cdot -\frac{1}{1-t} \cdot \left(\frac{2(1-\rho_j)^{1-t}}{(1-\rho_j)^{1-t} + (1+\rho_j)^{1-t}} - 1\right)$$

$$= -\frac{1}{R_j} \cdot \log_t\left(\frac{1-\rho_j}{M_{1-t}(1-\rho_j, 1+\rho_j)}\right),$$

where $M_q(a,b) \doteq ((a^q + b^q)/2)^{1/q}$ is the power mean with exponent $q$. We now check (56).

**Lemma H.** *For any $t \in \mathbb{R}$, let*

$$\mu_j \doteq -\frac{1}{R_j} \cdot \log_t\left(\frac{1-\rho_j}{M_{1-t}(1-\rho_j, 1+\rho_j)}\right). \tag{57}$$

*Then $|\mu_j| \leqslant 1/(R_j|1-t|)$.*

*Proof.* Equivalently, we must show

$$\left|\log_t\left(\frac{1-z}{M_{1-t}(1-z, 1+z)}\right)\right| \leqslant \frac{1}{|1-t|}, \forall z \in [-1,1],$$

which is equivalent to showing

$$\left|\frac{2(1-z)^{1-t}}{(1-z)^{1-t} + (1+z)^{1-t}} - 1\right| \left(= \left|\frac{1 - \left(\frac{1+z}{1-z}\right)^{1-t}}{1 + \left(\frac{1+z}{1-z}\right)^{1-t}}\right|\right) \leqslant 1, \forall z \in [-1,1].$$

Define function $f(z,t) \doteq (1 - z^{1-t})/(1 + z^{1-t})$ over $\mathbb{R}_{\geqslant 0} \times \mathbb{R}$: it is easy to check that for $t \leqslant 1$, $f(z,t) \in [-1,1]$, and the symmetry $f(z,t) = -f(z, 2-t)$ also allows to conclude that for $t \geqslant 1$, $f(z,t) \in [-1,1]$. This ends the proof of Lemma H. □

For the expression of $\mu_j$ in (57), we get from (55) the upperbound on $Z_{tj}^{2-t}(\mu_j)$:

$$Z_{tj}^{2-t}(\mu_j) \leqslant \frac{1 + m_j^\dagger q_j^{\dagger 2-t}}{2} \cdot \left((1+\rho_j) \cdot \exp_t^{2-t}(-\mu_j R_j) + (1-\rho_j) \cdot \exp_t^{2-t}(\mu_j R_j)\right)$$

$$= \frac{1 + m_j^\dagger q_j^{\dagger 2-t}}{2} \cdot \left(\frac{(1+\rho_j)(1-\rho_j)^{2-t}}{M_{1-t}^{2-t}(1-\rho_j, 1+\rho_j)} + \frac{(1-\rho_j)(1+\rho_j)^{2-t}}{M_{1-t}^{2-t}(1-\rho_j, 1+\rho_j)}\right)$$

$$= \left(1 + m_j^\dagger q_j^{\dagger 2-t}\right) \cdot \frac{(1-\rho_j^2) M_{1-t}^{1-t}(1-\rho_j, 1+\rho_j)}{M_{1-t}^{2-t}(1-\rho_j, 1+\rho_j)}$$

$$= \left(1 + m_j^\dagger q_j^{\dagger 2-t}\right) \cdot \frac{1-\rho_j^2}{M_{1-t}(1-\rho_j, 1+\rho_j)}.$$

We conclude that for both sets of classifiers defined in Lemmata F and G, with the choice of $\mu_j$ in (57), we get

$$\frac{1}{m} \cdot \sum_{i \in [m]} [\![\text{sign}(H(\boldsymbol{x}_i)) \neq y_i]\!] \leqslant \prod_{j=1}^{J} \left(1 + m_j^\dagger q_j^{\dagger 2-t}\right) \cdot \frac{1-\rho_j^2}{M_{1-t}(1-\rho_j, 1+\rho_j)}, \forall H \in \{H_J, H_J^{(1/1-t)}\}.$$

To complete the proof of Theorem 2, we just need to elicit the best $R_j$ (51) and $q_j^\dagger$ (52); looking at their constraints suggests

$$R_j \doteq \max_{i \notin [m]_j^\dagger} \frac{|y_i h_j(\boldsymbol{x}_i)|}{q_{ji}^{1-t}},$$

$$q_j^\dagger \doteq \frac{\max_{i \in [m]_j^\dagger} |y_i h_j(\boldsymbol{x}_i)|^{1/(1-t)}}{R_j^{1/(1-t)}}.$$

This completes the proof of Theorem 2. We complete the proof by two Lemmata of additional useful results in the context of Algorithm $t$-ADABOOST, and finally an important remark on the interpretation of Theorem 2.

**Lemma I.** *The following holds true: (i) $\rho_j \in [-1, 1]$; (ii) if, among indexes not in $[m]_j^\dagger$, there exists at least one index with $u_{ji} > 0$ and one index with $u_{ji} < 0$, then for any $\mu \neq 0$, $Z_{tj}^{2-t}(\mu) > 0$ in (50) (in words, the new weigh vector $\boldsymbol{q}_{j+1}$ cannot be the null vector before normalization).*

*Proof.* To show (i) for $\rho_j \leqslant 1$, we write (using $u_{ji} \doteq y_i h_j(\boldsymbol{x}_i), \forall i \in [m]$ for short),

$$
\begin{aligned}
(1 + m_j^\dagger q_j^{\dagger 2-t}) R_j \cdot \rho_j &= \sum_{i \in [m]} q'_{ji} u_{ji} \\
&\leqslant \sum_{i \in [m]} q'_{ji} |u_{ji}| \\
&= \sum_{i \in [m]_j^+} q_{ji}^{2-t} \cdot \frac{|u_{ji}|}{q_{ji}^{1-t}} + q_j^{\dagger 2-t} \cdot \frac{\sum_{i \in [m]_j^\dagger} |u_{ji}|}{q_j^{\dagger 1-t}} \\
&\leqslant R_j \cdot \underbrace{\sum_{i \in [m]_j^+} q_{ji}^{2-t}}_{=1} + q_j^{\dagger 2-t} \cdot \frac{R_j \sum_{i \in [m]_j^\dagger} |u_{ji}|}{\max_{i \in [m]_j^\dagger} |u_{ji}|} \\
&\leqslant R_j + q_j^{\dagger 2-t} m_j^\dagger R_j = (1 + m_j^\dagger q_j^{\dagger 2-t}) R_j,
\end{aligned}
$$

showing $\rho_j \leqslant 1$. Showing $\rho_j \geqslant -1$ proceeds in the same way. Property (ii) is trivial. $\qquad\square$

**Lemma J.**

$$K_t(z) \leqslant \exp\left(-\left(1 - \frac{t}{2}\right) \cdot z^2\right).$$

*Proof.* We remark that for $t \in [0, 1), z \geqslant 0$, $K'_t(z)$ is concave and $K''_t(0) = -(2 - t)$, so $K'_t(z) \leqslant -(2-t)z, \forall z \geqslant 0$, from which it follows by integration

$$K_t(z) \leqslant 1 - \left(1 - \frac{t}{2}\right) \cdot z^2,$$

and since $1 - z \leqslant \exp(-z)$, we get the statement of the Lemma. $\qquad\square$

**Remark 1.** *The interpretation of Theorem 2 for $t < 1$ are simplified to the case where there is no weight switching, i.e. $m_j^\dagger = 0, \forall j$. While we have never observed weight switching in our experiments – perhaps owing to the fact that we did never boost for a very long number of iterations or just because our weak classifiers, decision trees, were in fact not so weak –, it is interesting, from a theoretical standpoint, to comment on convergence when this happens. Let $Q_j \doteq 1 + m_j^\dagger (q_j^\dagger)^{2-t}$ and $\tilde{\rho}_j \doteq Q_j \rho_j$ (Notations from Theorem 2). We note that $\tilde{\rho}_j \approx \beta \cdot \mathbb{E}_{\boldsymbol{p}_j}[y_i h_j(\boldsymbol{x}_i)]$, where $\boldsymbol{p}_j$ lives on the simplex and $|yh| \leqslant 1, \beta \leqslant 1$. Using Lemma J and (12) (main file), to keep geometric convergence, it is roughly sufficient that $Q_j \log Q_j \leqslant (\tilde{\rho}_j)^2/(2t^*)$. Since $q_j^\dagger$ is homogeneous to a tempered weight, one would expect in general $m_j^\dagger (q_j^\dagger)^{2-t} \ll 1$, so using the Taylor approximation $Q_j \log Q_j \approx -1 + Q_j$, one gets the refined sufficient condition for geometric convergence*

$$m_j^\dagger (q_j^\dagger)^{2-t} \leqslant (\tilde{\rho}_j)^2/(2t^*) = O((\tilde{\rho}_j)^2).$$

*What does that imply? We have two cases:*

- *If this holds, then we have geometric convergence;*

- *if it does not hold, then for a "large" number of training examples, we must have $q_{ji} = 0$ which, because of the formula for $\boldsymbol{q}$ (8) implies that all these examples receive the right class with a sufficiently large margin. Breaking geometric convergence in this case is not an issue: we already have a good ensemble.*

## II.3  Proof of Theorem 3

Starting from the proof of Theorem 2, we indicate the additional steps to get to the proof of Theorem 3. The key is to remark that our margin formulation has the following logical convenience:

$$
\begin{aligned}
\llbracket \nu_t((\boldsymbol{x}_i, y_i), H) \leqslant \theta \rrbracket &= \llbracket -yH(\boldsymbol{x}) + \log_t\left(\frac{1+\theta}{1-\theta}\right) - (1-t)yH(\boldsymbol{x})\log_t\left(\frac{1+\theta}{1-\theta}\right) \geqslant 0 \rrbracket \\
&= \llbracket (-yH(\boldsymbol{x})) \oplus_t \log_t\left(\frac{1+\theta}{1-\theta}\right) \geqslant 0 \rrbracket.
\end{aligned}
$$

We then remark that since $\llbracket z \geqslant 0 \rrbracket \leqslant \exp_t^{2-t}(z)$, we get

$$
\begin{aligned}
\llbracket \nu_t((\boldsymbol{x}_i, y_i), H) \leqslant \theta \rrbracket &\leqslant \exp_t^{2-t}\left( (-yH(\boldsymbol{x})) \oplus_t \log_t\left(\frac{1+\theta}{1-\theta}\right) \right) \\
&= \exp_t^{2-t}\left( \log_t\left(\frac{1+\theta}{1-\theta}\right) \right) \cdot \exp_t^{2-t}(-yH(\boldsymbol{x})) \\
&= \left(\frac{1+\theta}{1-\theta}\right)^{2-t} \cdot \exp_t^{2-t}(-yH(\boldsymbol{x})).
\end{aligned}
$$

We then just have to branch to (44), replacing the $\llbracket \mathrm{sign}(H_J(\boldsymbol{x}_i)) \neq y_i \rrbracket$s by $\llbracket \nu_t((\boldsymbol{x}_i, y_i), H) \leqslant \theta \rrbracket$, which yields in lieu of (46) the sought inequality:

$$
F_{t,\theta}(H, \mathcal{S}) \leqslant \left(\frac{1+\theta}{1-\theta}\right)^{2-t} \prod_{j=1}^{J} Z_{tj}^{2-t}. \tag{58}
$$

## II.4  Proof of Theorem 4

The proof proceeds in three parts. Part **(A)** makes a brief recall on encoding linear classifiers with decision trees. Part **(B)** solves (6) in MF, *i.e.* finds boosting's leveraging coefficients as solution of:

$$
\boldsymbol{q}(\mu)^\top \boldsymbol{u} = 0. \tag{59}
$$

we then simplify the loss obtained and elicit the conditional Bayes risk of the tempered loss, *i.e.* (20) in MF. Part **(C)** elicits the partial losses and shows properness and related properties.

**Part (A): encoding linear models with a tree architecture**  We use the reduction trick of **(author?)** [16] to design a decision tree (DT) boosting procedure, find out the (concave) loss equivalently minimized, just like in classical top-down DT induction algorithms [6]. The trick is simple: a DT can be thought of as a set of constant linear classifiers. The prediction is the sum of predictions put at all nodes. Boosting fits those predictions at the nodes and percolating those to leaves gets a standard DT with real predictions at the leaves. Figure 2 provides a detailed description of the procedure. Let $\lambda$ denote a leaf node of the current tree $H$, with $H_\lambda \in \mathbb{R}$ the function it implements for leaf $\lambda$. If $\mathrm{parent}(\lambda)$ denotes its parent node (assuming wlog it is not the root node), we have

$$
H_\lambda \doteq H_{\mathrm{parent}(\lambda)} + \mu_\lambda h_\lambda, \tag{60}
$$

**Part (B): eliciting the Bayes risk of the tempered loss**  With our simple classifiers at hand, the tempered exponential loss $Z_{tj}^{2-t}$ in (14) (MF) can be simplified to loss

$$
\begin{aligned}
L(H) &\doteq \sum_i \exp_t^{2-t}\left( \log_t q_{1i} - y_i H_{\lambda(\boldsymbol{x}_i)} \right) \\
&= \sum_{\lambda \in \Lambda(H)} m_\lambda^+ \exp_t^{2-t}\left( \log_t q_{1i} - H_\lambda \right) + m_\lambda^- \exp_t^{2-t}\left( \log_t q_{1i} + H_\lambda \right), \tag{61}
\end{aligned}
$$

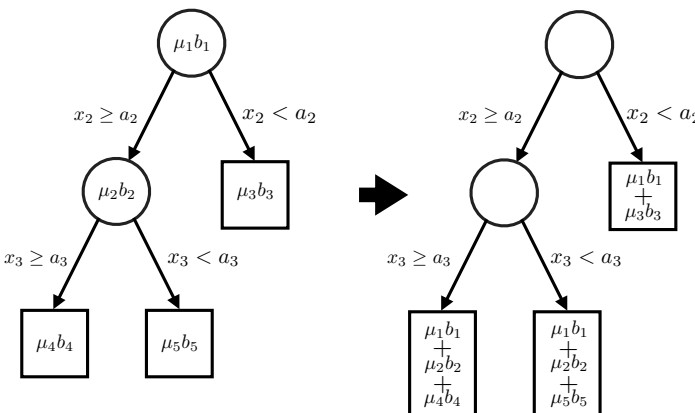

Figure 2: The weak learner provides weak hypotheses of the form $[\![x_k \geqslant a_j]\!] \cdot b_j$. From the boosting standpoint, this weak hypothesis is "as good as" the weak hypothesis $\overline{h}_j(\boldsymbol{x}) \doteq [\![x_k < a_j]\!] \cdot -b_j$. The predicates of both are used to craft a split, *e.g.* for the root (in our depiction, $b_3 = -b_2$) and then solving (59) provides the leveraging coefficients $\mu_.$. We then repeat this for as many splits as necessary. At the end, we can "percolate" nodes reals towards the leaves below and get an equivalent classifier that resembles a decision tree (right). See [16] for further details.

where $\lambda(\boldsymbol{x})$ is the leaf reached by observation $\boldsymbol{x}$ and $\lambda(H)$ its set of leaf nodes of $H$, and $H_\lambda$ sums all relevant values in (60). Also, $m_\lambda^+, m_\lambda^-$ denote the cardinal of positive and negative examples at $\lambda$ and $p_\lambda \doteq m_\lambda^+/(m_\lambda^+ + m_\lambda^-)$ the local proportion of positive examples at $\lambda$, and finally $r_\lambda \doteq (m_\lambda^+ + m_\lambda^-)/m$ the total proportion of examples reaching $\lambda$.

**Theorem A.** *If we compute $\mu_\lambda$ the solution of* (59)*, we end up with the prediction $H_\lambda$:*

$$H_\lambda = \frac{q_{1i}^{1-t}}{1-t} \cdot \frac{\left(\frac{m_\lambda^+}{m_\lambda^-}\right)^{1-t} - 1}{\left(\frac{m_\lambda^+}{m_\lambda^-}\right)^{1-t} + 1} \tag{62}$$

$$= \frac{q_{1i}^{1-t}}{1-t} \cdot \frac{p_\lambda^{1-t} - (1-p_\lambda)^{1-t}}{p_\lambda^{1-t} + (1-p_\lambda)^{1-t}}, \tag{63}$$

*and the loss of the decision tree equals:*

$$L(H) = \sum_{\lambda \in \Lambda(H)} r_\lambda \cdot \frac{2p_\lambda(1-p_\lambda)}{M_{1-t}(p_\lambda, 1-p_\lambda)}, \tag{64}$$

$$= \mathbb{E}_\lambda[\underline{L}^{(t)}(p_\lambda)]. \tag{65}$$

*Proof.* To compute $\mu_\lambda$, (6) is reduced to the examples reaching $\lambda$, that is, it simplifies to

$$m_\lambda^+ \exp_t \left(\log_t q_{1i} - H_{\text{parent}(\lambda)} - R_\lambda \mu_\lambda h_\lambda\right) = m_\lambda^- \exp_t \left(\log_t q_{1i} + H_{\text{parent}(\lambda)} + R_\lambda \mu_\lambda h_\lambda\right) \tag{66}$$

that we solve for $\mu_\lambda$. Equivalently,

$$\frac{\exp_t \left(\log_t q_{1i} + H_{\text{parent}(\lambda)} + R_\lambda \mu_\lambda h_\lambda\right)}{\exp_t \left(\log_t q_{1i} - H_{\text{parent}(\lambda)} - R_\lambda \mu_\lambda h_\lambda\right)} = \frac{m_\lambda^+}{m_\lambda^-},$$

or, using $\exp_t(u)/\exp_t(v) = \exp_t(u \ominus_t v)$,

$$\frac{2H_{\text{parent}(\lambda)} + 2R_\lambda \mu_\lambda h_\lambda}{1 + (1-t)(\log_t q_{1i} - H_{\text{parent}(\lambda)} - R_\lambda \mu_\lambda h_\lambda)} = \log_t \left(\frac{m_\lambda^+}{m_\lambda^-}\right),$$

after reorganizing:

$$R_\lambda \mu_\lambda h_\lambda = \frac{(1 + (1-t)(\log_t q_{1i} - H_{\text{parent}(\lambda)})) \cdot \log_t \left(\frac{m_\lambda^+}{m_\lambda^-}\right) - 2H_{\text{parent}(\lambda)}}{2 + (1-t)\log_t \left(\frac{m_\lambda^+}{m_\lambda^-}\right)},$$

which yields the prediction at $\lambda$:

$$H_\lambda \;=\; H_{\mathrm{parent}(\lambda)} + \frac{\left(1 + (1-t)(\log_t q_{1i} - H_{\mathrm{parent}(\lambda)})\right)\cdot\log_t\left(\frac{m_\lambda^+}{m_\lambda^-}\right) - 2H_{\mathrm{parent}(\lambda)}}{2 + (1-t)\log_t\left(\frac{m_\lambda^+}{m_\lambda^-}\right)} \tag{67}$$

$$=\; \frac{\left(1 + (1-t)\log_t q_{1i}\right)\cdot\log_t\left(\frac{m_\lambda^+}{m_\lambda^-}\right)}{2 + (1-t)\log_t\left(\frac{m_\lambda^+}{m_\lambda^-}\right)} \tag{68}$$

$$=\; q_{1i}^{1-t}\cdot\frac{\log_t\left(\frac{m_\lambda^+}{m_\lambda^-}\right)}{2 + (1-t)\log_t\left(\frac{m_\lambda^+}{m_\lambda^-}\right)} \tag{69}$$

$$=\; \frac{q_{1i}^{1-t}}{1-t}\cdot\frac{\left(\frac{m_\lambda^+}{m_\lambda^-}\right)^{1-t} - 1}{\left(\frac{m_\lambda^+}{m_\lambda^-}\right)^{1-t} + 1} \tag{70}$$

$$=\; \frac{q_{1i}^{1-t}}{1-t}\cdot\frac{p_\lambda^{1-t} - (1-p_\lambda)^{1-t}}{p_\lambda^{1-t} + (1-p_\lambda)^{1-t}}. \tag{71}$$

We plug $H_\lambda$ back in the loss for all leaves and get, using $q_{1i} = 1/m^{1/(2-t)}$:

$$L(H) \;=\; \sum_{\lambda\in\Lambda(H)} \left\{ \begin{array}{l} m_\lambda^+ \exp_t^{2-t}\left(\log_t q_{1i} - \frac{q_{1i}^{1-t}}{1-t}\cdot\frac{p_\lambda^{1-t}-(1-p_\lambda)^{1-t}}{p_\lambda^{1-t}+(1-p_\lambda)^{1-t}}\right) \\ +m_\lambda^- \exp_t^{2-t}\left(\log_t q_{1i} + \frac{q_{1i}^{1-t}}{1-t}\cdot\frac{p_\lambda^{1-t}-(1-p_\lambda)^{1-t}}{p_\lambda^{1-t}+(1-p_\lambda)^{1-t}}\right) \end{array} \right\}. \tag{72}$$

We simplify. First,

$$m_\lambda^+ \exp_t^{2-t}\left(\log_t q_{1i} - \frac{q_{1i}^{1-t}}{1-t}\cdot\frac{p_\lambda^{1-t} - (1-p_\lambda)^{1-t}}{p_\lambda^{1-t} + (1-p_\lambda)^{1-t}}\right)$$

$$=\; m_\lambda^+ \left[q_{1i}^{1-t}\cdot\left(1 - \frac{p_\lambda^{1-t} - (1-p_\lambda)^{1-t}}{p_\lambda^{1-t} + (1-p_\lambda)^{1-t}}\right)\right]_+^{\frac{2-t}{1-t}}$$

$$=\; \frac{m_\lambda^+}{m}\cdot\left[\frac{2(1-p_\lambda)^{1-t}}{p_\lambda^{1-t} + (1-p_\lambda)^{1-t}}\right]_+^{\frac{2-t}{1-t}} \tag{73}$$

$$=\; \frac{m_\lambda^+}{m}\cdot\left(\frac{1-p_\lambda}{M_{1-t}(p_\lambda, 1-p_\lambda)}\right)^{2-t}, \tag{74}$$

and then

$$m_\lambda^- \exp_t^{2-t}\left(\log_t q_{1i} + \frac{q_{1i}^{1-t}}{1-t}\cdot\frac{p_\lambda^{1-t} - (1-p_\lambda)^{1-t}}{p_\lambda^{1-t} + (1-p_\lambda)^{1-t}}\right)$$

$$=\; m_\lambda^- \left[q_{1i}^{1-t}\cdot\left(1 + \frac{p_\lambda^{1-t} - (1-p_\lambda)^{1-t}}{p_\lambda^{1-t} + (1-p_\lambda)^{1-t}}\right)\right]_+^{\frac{2-t}{1-t}}$$

$$=\; \frac{m_\lambda^-}{m}\cdot\left[\frac{2p_\lambda^{1-t}}{p_\lambda^{1-t} + (1-p_\lambda)^{1-t}}\right]_+^{\frac{2-t}{1-t}} \tag{75}$$

$$=\; \frac{m_\lambda^-}{m}\cdot\left(\frac{p_\lambda}{M_{1-t}(p_\lambda, 1-p_\lambda)}\right)^{2-t}, \tag{76}$$

and we can simplify the loss,

$$L(H) \;=\; \sum_{\lambda \in \Lambda(H)} r_\lambda p_\lambda \left( \frac{1-p_\lambda}{M_{1-t}(p_\lambda, 1-p_\lambda)} \right)^{2-t} + r_\lambda(1-p_\lambda) \left( \frac{p_\lambda}{M_{1-t}(p_\lambda, 1-p_\lambda)} \right)^{2-t} \quad (77)$$

$$= \sum_{\lambda \in \Lambda(H)} r_\lambda \cdot \frac{p_\lambda(1-p_\lambda)^{2-t} + (1-p_\lambda)p_\lambda^{2-t}}{M_{1-t}^{2-t}(p_\lambda, 1-p_\lambda)} \quad (78)$$

$$= \sum_{\lambda \in \Lambda(H)} r_\lambda \cdot \frac{p_\lambda(1-p_\lambda) \cdot (p_\lambda^{1-t} + (1-p_\lambda)^{1-t})}{M_{1-t}^{2-t}(p_\lambda, 1-p_\lambda)} \quad (79)$$

$$= \sum_{\lambda \in \Lambda(H)} r_\lambda \cdot \frac{2p_\lambda(1-p_\lambda) \cdot M_{1-t}^{1-t}(p_\lambda, 1-p_\lambda)}{M_{1-t}^{2-t}(p_\lambda, 1-p_\lambda)} \quad (80)$$

$$= \sum_{\lambda \in \Lambda(H)} r_\lambda \cdot \frac{2p_\lambda(1-p_\lambda)}{M_{1-t}(p_\lambda, 1-p_\lambda)}, \quad (81)$$

as claimed. This ends the proof of Theorem A. □

**Part (C): partial losses and their properties**  The proof relies on the following Theorem. We recall that a loss is symmetric iff its partial losses satisfy $\ell_1(u) = \ell_{-1}(1-u), \forall u \in [0,1]$ [20] and differentiable iff its partial losses are differentiable.

**Theorem B.** *Suppose $t < 2$. A set of partial losses having the conditional Bayes risk $\underline{L}^{(t)}$ in (65) are*

$$\ell_1^{(t)}(u) \doteq \left( \frac{1-u}{M_{1-t}(u, 1-u)} \right)^{2-t} \quad , \quad \ell_{-1}^{(t)}(u) \doteq \ell_1^{(t)}(1-u). \quad (82)$$

*The tempered loss is then symmetric and differentiable. It is strictly proper for any $t \in (-\infty, 2)$ and proper for $t = -\infty$.*

*Proof.* Symmetry and differentiability are straightforward. To check strict properness, we analyze the cases for $t \neq 1$ (otherwise, it is Matusita's loss, and thus strictly proper), we compute the solution $u$ to

$$\frac{\partial}{\partial u} L(u, v) \;=\; 0. \quad (83)$$

To this end, let $N(u) \doteq v(1-u)^{2-t} + (1-v)u^{2-t}$ and the $q$-sum

$$S_q(a, b) \;\doteq\; (a^q + b^q)^{1/q} = 2^{1/q} \cdot M_q(a, b). \quad (84)$$

We also let $D(u) \doteq S_{1-t}^{2-t}(u, 1-u)$. Noting $L^{(t)}(u, v) = 2^{\frac{2-t}{1-t}} \cdot N(u)/D(u)$ and $D(u) \neq 0, \forall u \in [0,1]$, the set of solutions of (83) are the set of solutions to $N'(u)D(u) = N(u)D'(u)$, which boils down, after simplification, to

$$((1-v)u^{1-t} - v(1-u)^{1-t})S_{1-t}^{2-t}(u, 1-u)$$
$$= (v(1-u)^{2-t} + (1-v)u^{2-t})(u^{-t} - (1-u)^{-t})S_{1-t}(u, 1-u),$$

developing and simplifying yields a first simplified expression $(1 - 2v)(u(1-u))^{1-t} = vu^{-1}(1-u)^{2-t} - (1-v)(1-u)^{-t}u^{2-t}$, which, after reorganising to isolate expressions depending on $v$, yields

$$(u(1-u))^{1-t} + (1-u)^{-t}u^{2-t} \;=\; v \cdot \left( u^{-t}(1-u)^{2-t} + (1-u)^{-t}u^{2-t} + 2(u(1-u))^{1-t} \right) \quad (85)$$

Assuming $v \in (0,1)$, we multiply by $(u(1-u))^{1-t}$ (we shall check $u \in (0,1)$) and simplify, which yields $u(1-u) + u^2 = v((1-u)^2 + u^2 + 2u(1-u))$, and indeed yields

$$u \;=\; v, \quad (86)$$

and we check from (85) that if $v = 0$ (resp. $v = 1$), then necessarily $u = 0$ (resp. $u = 1$). To complete the proof, using the previous derivations, we can then simplify

$$\frac{\partial}{\partial u} L(u, v) \;=\; (2 - t) \cdot 2^{\frac{2-t}{1-t}} \cdot \frac{u - v}{(u(1-u))^t \cdot S_{1-t}^{3-2t}(u, 1-u)}, \quad (87)$$

which shows that if $2 - t > 0$ but $t \neq -\infty$, $u = v$ is a strict minimum of the pointwise conditional risk, completing the proof for strict properness. Strict properness is sufficient to show by a simple computation that $\underline{L}^{(t)}$ is (65). For $t = -\infty$, we pass to the limit and use the fact that we can also write

$$\ell_1^{(t)}(u) \quad = \quad \frac{1}{M_{1-t*}\left(1, \left(\frac{u}{1-u}\right)^{\frac{1}{t*}}\right)} \qquad \text{(we recall } t^* \doteq 1/(2-t)) \tag{88}$$

$t \to -\infty$ is equivalent to $t^* \to 0^+$. If $u < 1/2$, $u/(1-u) < 1$ and so we see that

$$\lim_{t* \to 0^+} M_{1-t*}\left(1, \left(\frac{u}{1-u}\right)^{\frac{1}{t*}}\right) \quad = \quad \frac{1}{2},$$

because $M_1$ is the arithmetic mean. When $u > 1/2$, $u/(1-u) > 1$ and so this time

$$\lim_{t* \to 0^+} M_{1-t*}\left(1, \left(\frac{u}{1-u}\right)^{\frac{1}{t*}}\right) \quad = \quad +\infty.$$

Hence,

$$\ell_1^{(-\infty)}(u) \quad = \quad 2 \cdot \llbracket u \leqslant 1/2 \rrbracket, \tag{89}$$

which is (twice) the partial loss of the 0/1 loss [26]. $\qquad \square$

This ends the proof of Theorem 4.

| Domain | Source | $m$ | $d$ | |
|--------|--------|-----|-----|---|
| sonar | UCI | 208 | 60 | `https://archive.ics.uci.edu/ml/datasets/wine+quality` |
| winered | UCI | 1 599 | 12 | `https://archive.ics.uci.edu/ml/datasets/wine+quality` |
| abalone | UCI | 4 177 | 9 | `https://archive.ics.uci.edu/ml/datasets/abalone` |
| qsar | UCI | 1 055 | 41 | `https://archive.ics.uci.edu/ml/datasets/QSAR+biodegradation` |
| winewhite | UCI | 4 898 | 12 | `https://archive.ics.uci.edu/ml/datasets/wine+quality` |
| hillnonoise | UCI | 1 212 | 101 | `http://archive.ics.uci.edu/ml/datasets/hill-valley` |
| hillnoise | UCI | 1 212 | 101 | `http://archive.ics.uci.edu/ml/datasets/hill-valley` |
| eeg | UCI | 14 980 | 15 | `https://archive.ics.uci.edu/ml/datasets/EEG+Eye+State` |
| creditcard* | UCI | 14 599 | 24 | `https://archive.ics.uci.edu/ml/datasets/default+of+credit+card+clients` |
| adult | UCI | 32 561 | 15 | `https://archive.ics.uci.edu/ml/datasets/adult` |

Table A3: Public domains considered in our experiments ($m$ = total number of examples, $d$ = total number of example's features, including class), ordered in increasing $m \times d$ (see text). (*) first $m$ rows in the domain.

## III  Supplementary material on experiments

### III.1  Domains

Table A3 presents the 10 domains we used for our experiments.

### III.2  Implementation details and full set of experiments on linear combinations of decision trees

**Summary**  This Section depicts the full set of experiments summarized in Table 2 (MF), from Table A4 to Table A15. Tables are ordered in increasing size of the domain (Table A3). In all cases, up to $J = 20$ trees have been trained, of size 15 (total number of nodes, except the two biggest domains, for which the size is 5). For all datasets, except creditcard and adult, we have tested $t$ in the complete range, $t \in \{0.0, 0.2, 0.4, 0.6, 0.8, 0.9, 1.0, 1.1\}$ (the MF only reports results for $t \geqslant 0.6$), and in all cases, models both clipped and not clipped. For each dataset, we have set a 10-fold stratified cross-validation experiment, and report the averages for readability (Table 2 in MF gives the results of a Student paired $t$-test on error averages for comparison, limit $p$-val = 0.1). We also provide two examples of training error averages for domains hillnoise and hillnonoise (Tables A10 and A12).

**Implementation details of $t$-ADABOOST**  First, regarding file format, we only input a `.csv` file to $t$-ADABOOST. We do not specify a file with feature types as in ARFF files. $t$-ADABOOST recognizes the type of each feature from its column content and distinguishes two main types of features: numerical and categorical. The distinction is important to design the splits during decision tree induction: for numerical values, splits are midpoints between two successive observed values. For categorical, splits are partitions of the feature values in two non-empty subsets. Our implementation of $t$-ADABOOST (programmed in Java) makes it possible to choose $t$ not just in the range of values for which we have shown that boosting-compliant convergence is possible ($t \in [0, 1]$), but also $t > 1$. Because we thus implement ADABOOST ($t = 1$) but also for $t > 1$, weights can fairly easily become infinite, we have implemented a safe-check during training, counting the number of times the weights become infinite or zero (note that in this latter case, this really is a problem just for ADABOOST because in theory this should never happen unless the weak classifiers achieve perfect (or perfectly wrong) classification), but also making sure leveraging coefficients for classifiers do not become infinite for ADABOOST, a situation that can happen because of numerical approximations in encoding. In our experiments, we have observed that none of these problematic cases did occur (notice that this could not be the case if we were to boost for a large number of iterations). We have implemented algorithm $t$-ADABOOST exactly as specified in MF. The weak learner is implemented to train a decision tree in which the stopping criterion is the size of the tree reaching a user-fixed number of nodes. There is thus no pruning. Also, the top-down induction algorithm proceeds by iteratively picking the heaviest leaf in the tree and then choosing the split that minimizes the expected Bayes risk of the tempered loss, computing using the same $t$ values as for $t$-ADABOOST, and with the constraint to not get pure leaves (otherwise, the real prediction at the leaves, which relies on the

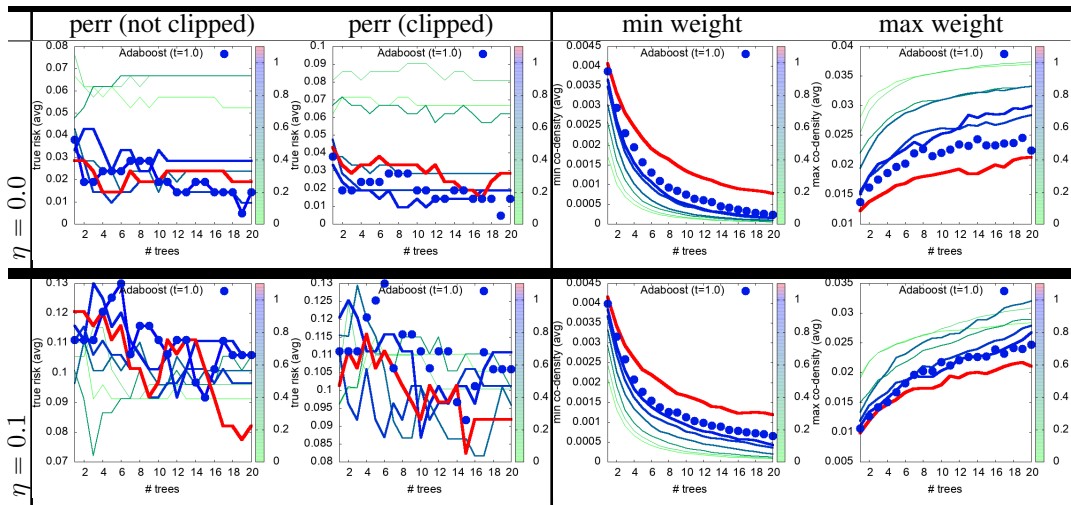

Table A4: Experiments on $t$-ADABOOST comparing with ADABOOST ($t = 1$, bullets) on domain sonar, when trained without noise ($\eta = 0.0$, top row) and with noise ($\eta = 0.1$, bottom row). Columns are, from left to right, the estimated true error of non-clipped and clipped models, and the min and max codensity weights. The set of $t$ values used is displayed in each plot with a colormap (right), and varying thickness of curves for an additional ease of reading (the thicker the curve, the larger $t$). ADABOOST's reference results are displayed with bullets. Averages shown for readability.

link of the loss, would be infinite for ADABOOST). In our implementation of decision-tree induction, when the number of possible splits exceeds a fixed number $S$ (currently, 2 000), we pick the best split in a subset of $S$ splits picked at random.

**Results** First, one may notice in several plots that the average test error increases with the number of trees. This turns out to be a sign of overfitting, as exemplified for domains hillnonoise and hillnoise, for which we provide the training curves. If we align the training curves at $T = 1$ (the value is different because the splitting criterion for training the tree is different), we notice that the experimental convergence on training is similar for all values of $t$ (Tables A10 and A12). The other key experimental result, already visible from Table 2 (MF), is that pretty much all tested values of $t$ are necessary to get the best results. One could be tempted to conclude that $t$ slightly smaller than 1.0 seems to be a good fit from Table 2 (MF), but the curves show that this is more a consequence of the Table being computed for $J = 20$ trees. The case of eeg illustrates this phenomenon best: while small $t$-values are clearly the best when there is no noise, the picture is completely reversed when there is training noise. Notice that this ordering is almost reversed on creditcard and adult: when there is noise, small values of $t$ tend to give better results. Hence, in addition to getting (i) a pruning mechanism that works for all instances of the tempered loss and (ii) a way to guess the right number of models in the ensemble, a good problem to investigate is in fact appropriately tuning $t$ in a domain-dependent way. Looking at all plots reveals that substantial gains could be obtained with an accurate procedure (over the strategy that would be to always pick a fixed $t$, *e.g.* $t = 1$).

### III.3   Supplementary experiments: learning with more trees / against more noise

We have performed some additional experiments on several domains, on which we have trained bigger ensembles and / or considered more noise levels than in the previous experiments, with the objective to see if / when overfitting happens and how performances degrade with noise as a function of $t$. Table A16 summarizes the results obtained. A few comments we can make based on those experiments are:

- overfitting can indeed happen (sonar for $\eta = 0.4$) but affects differently the algorithm depending on $t$ and whether clipped models are learned instead of regular linear models; results also display that tuning $t$ can also have the purpose of handling overfitting;

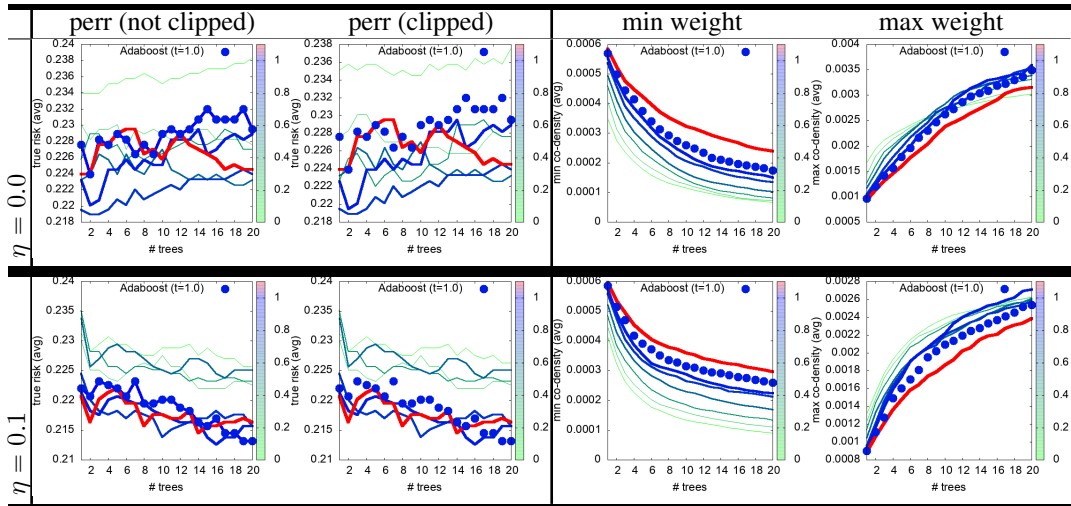

Table A5: Experiments on $t$-ADABOOST comparing with ADABOOST ($t = 1$, bullets) on domain `winered`. Conventions follow Table A4.

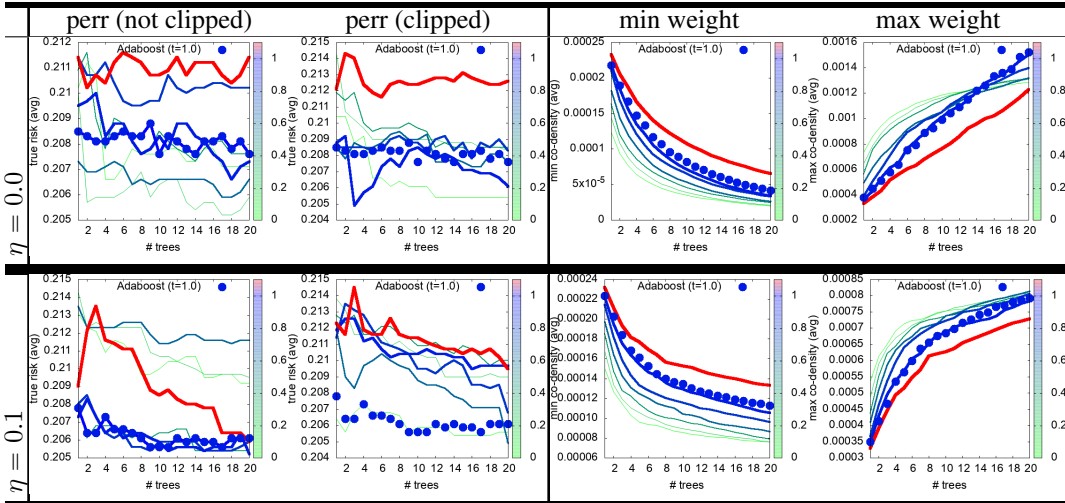

Table A6: Experiments on $t$-ADABOOST comparing with ADABOOST ($t = 1$, bullets) on domain `abalone`. Conventions follow Table A4.

- clamped models can be very useful to handle overfitting (`sonar` for $\eta = 0.4$, `qsar` for $\eta \geqslant 0.2$); this provides another justification to learn clamped models;
- the overall diversity of curves as a function of $t$ supports the idea that good strategies could in fact tune $t$ at training time and change its value with iterations.

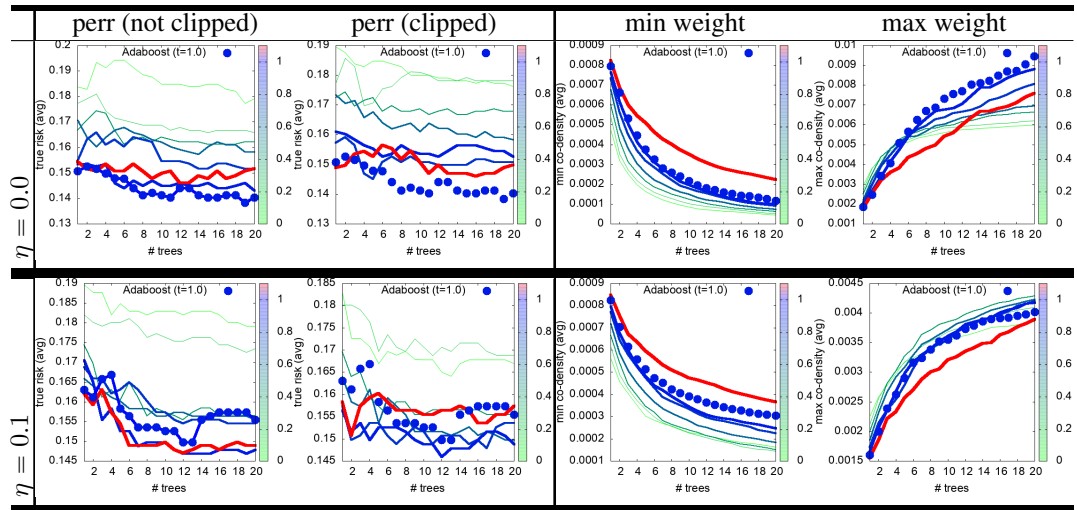

Table A7: Experiments on $t$-ADABOOST comparing with ADABOOST ($t = 1$, bullets) on domain qsar. Conventions follow Table A4.

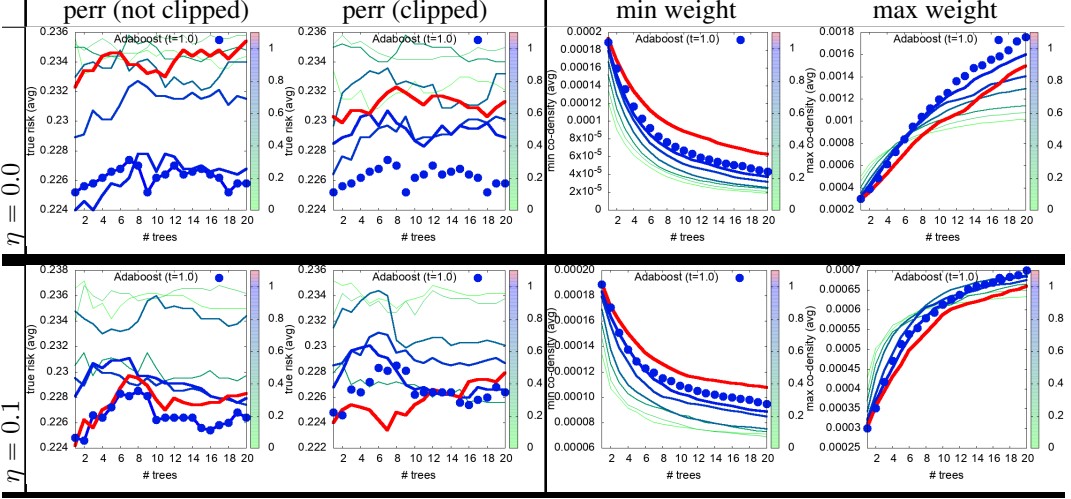

Table A8: Experiments on $t$-ADABOOST comparing with ADABOOST ($t = 1$, bullets) on domain winewhite. Conventions follow Table A4.

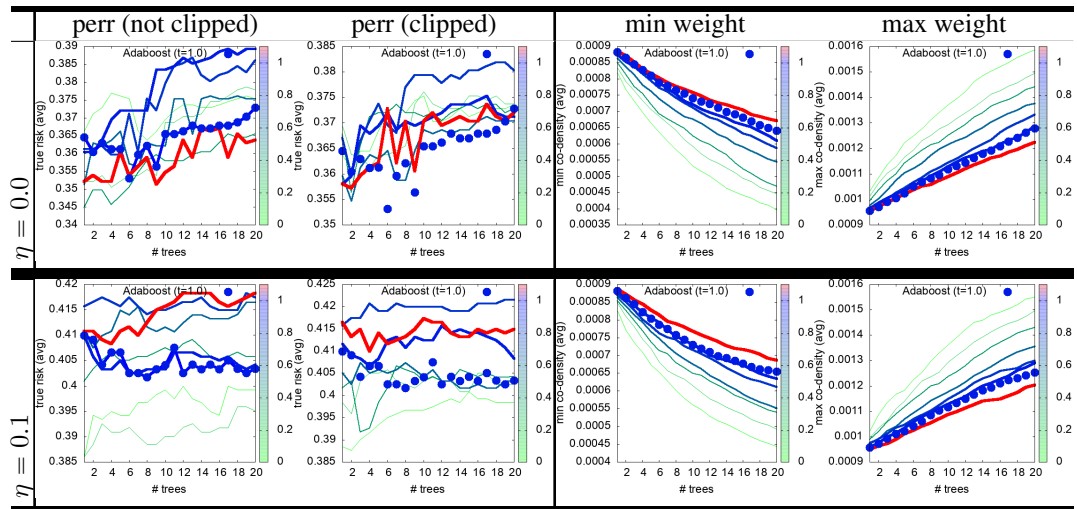

Table A9: Experiments on $t$-ADABOOST comparing with ADABOOST ($t = 1$, bullets) on domain `hillnonoise`. Conventions follow Table A4.

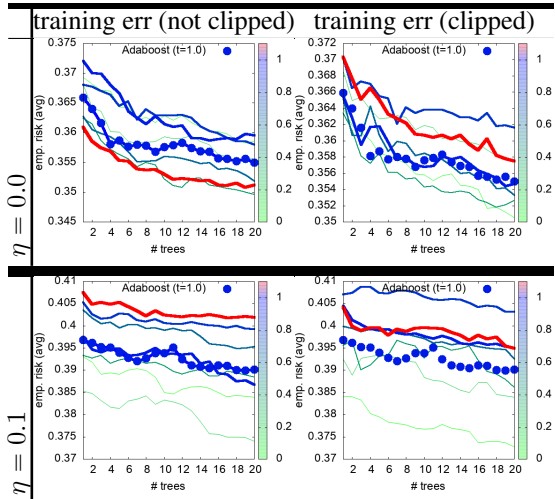

Table A10: Experiments on $t$-ADABOOST comparing with ADABOOST ($t = 1$, bullets) on domain `hillnonoise`: training errors displayed for all algorithms using conventions from Table A4. See text for details.

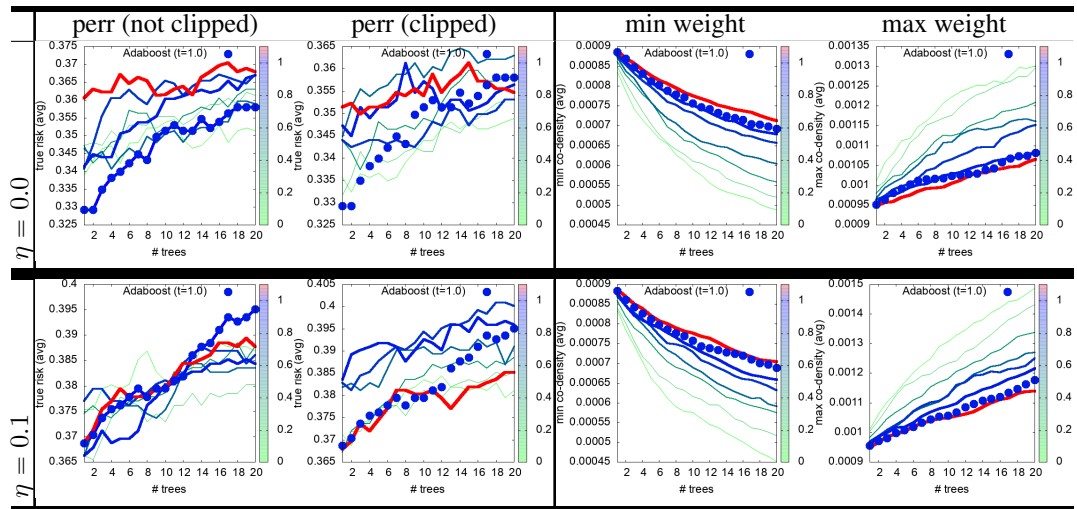

Table A11: Experiments on $t$-ADABOOST comparing with ADABOOST ($t = 1$, bullets) on domain `hillnoise`. Conventions follow Table A4.

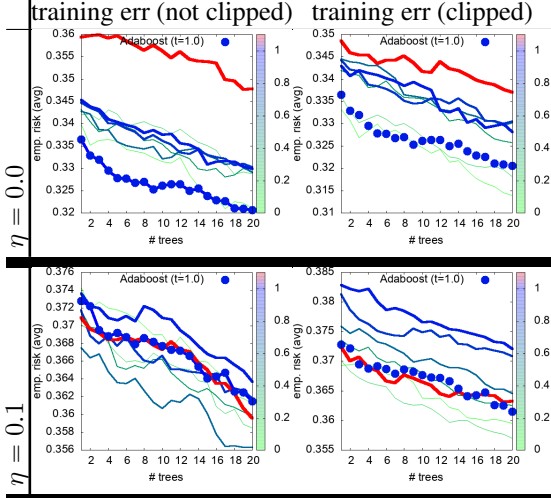

Table A12: Experiments on $t$-ADABOOST comparing with ADABOOST ($t = 1$, bullets) on domain `hillnoise`: training errors displayed for all algorithms using conventions from Table A4. See text for details.

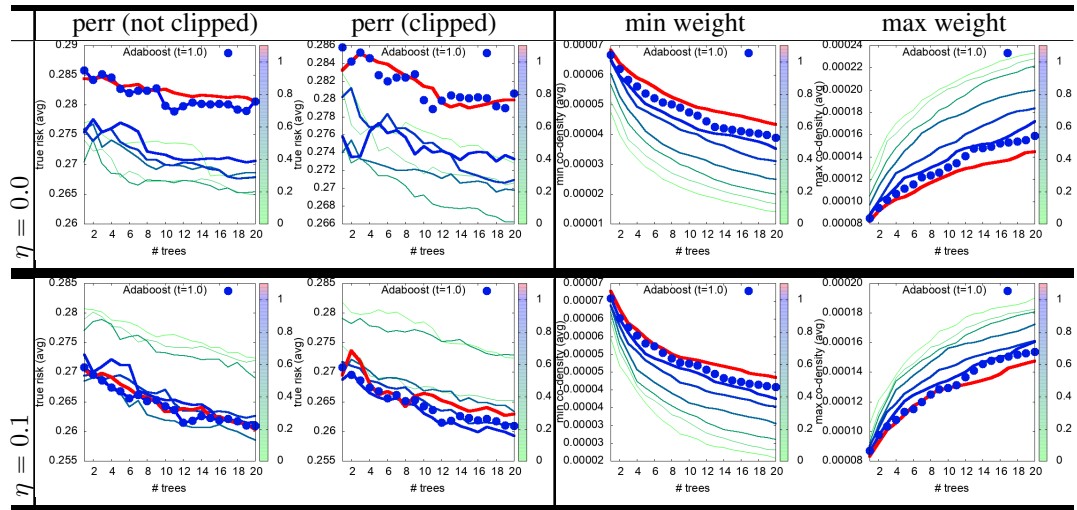

Table A13: Experiments on $t$-ADABOOST comparing with ADABOOST ($t = 1$, bullets) on domain `eeg`. Conventions follow Table A4.

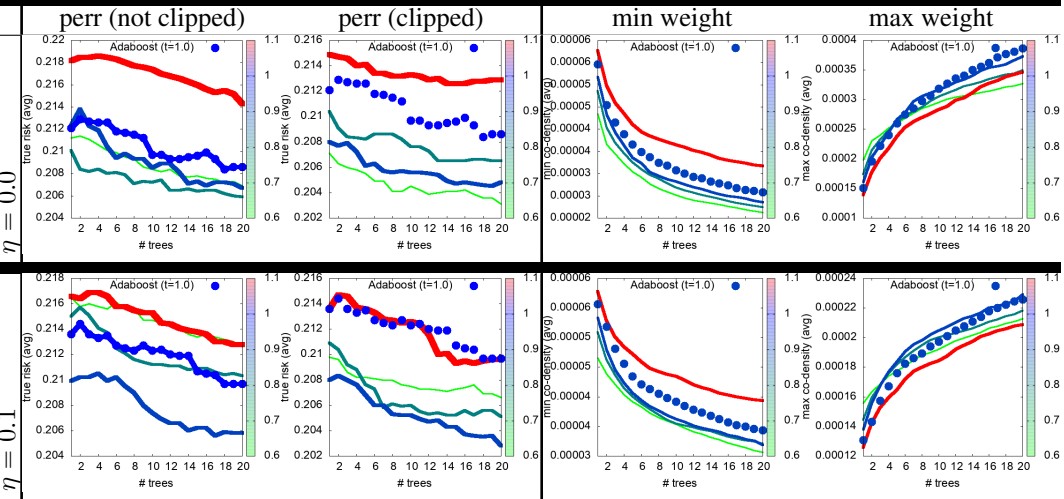

Table A14: Experiments on $t$-ADABOOST comparing with ADABOOST ($t = 1$, bullets) on domain `creditcard`. Conventions follow Table A4.

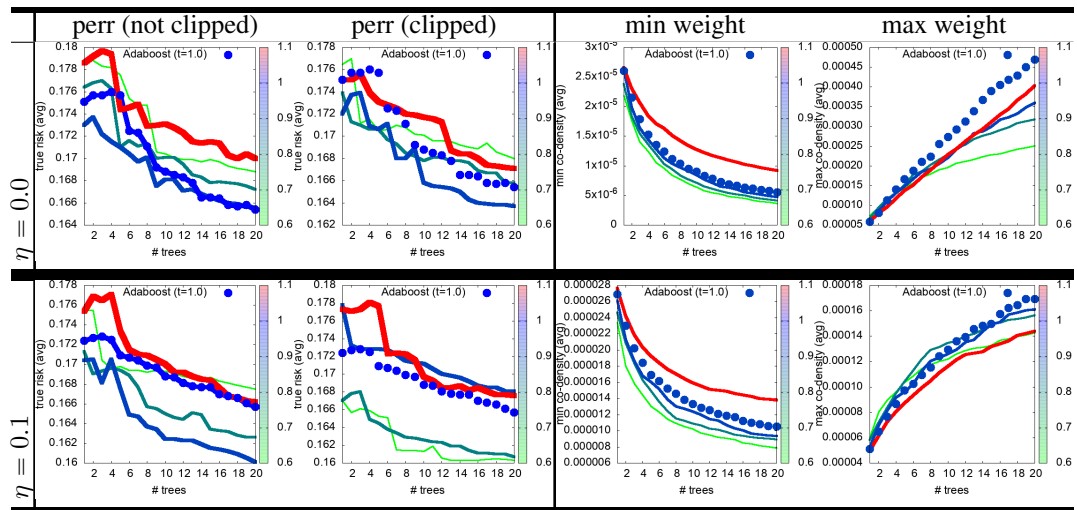

Table A15: Experiments on $t$-ADABOOST comparing with ADABOOST ($t = 1$, bullets) on domain adult. Conventions follow Table A4.

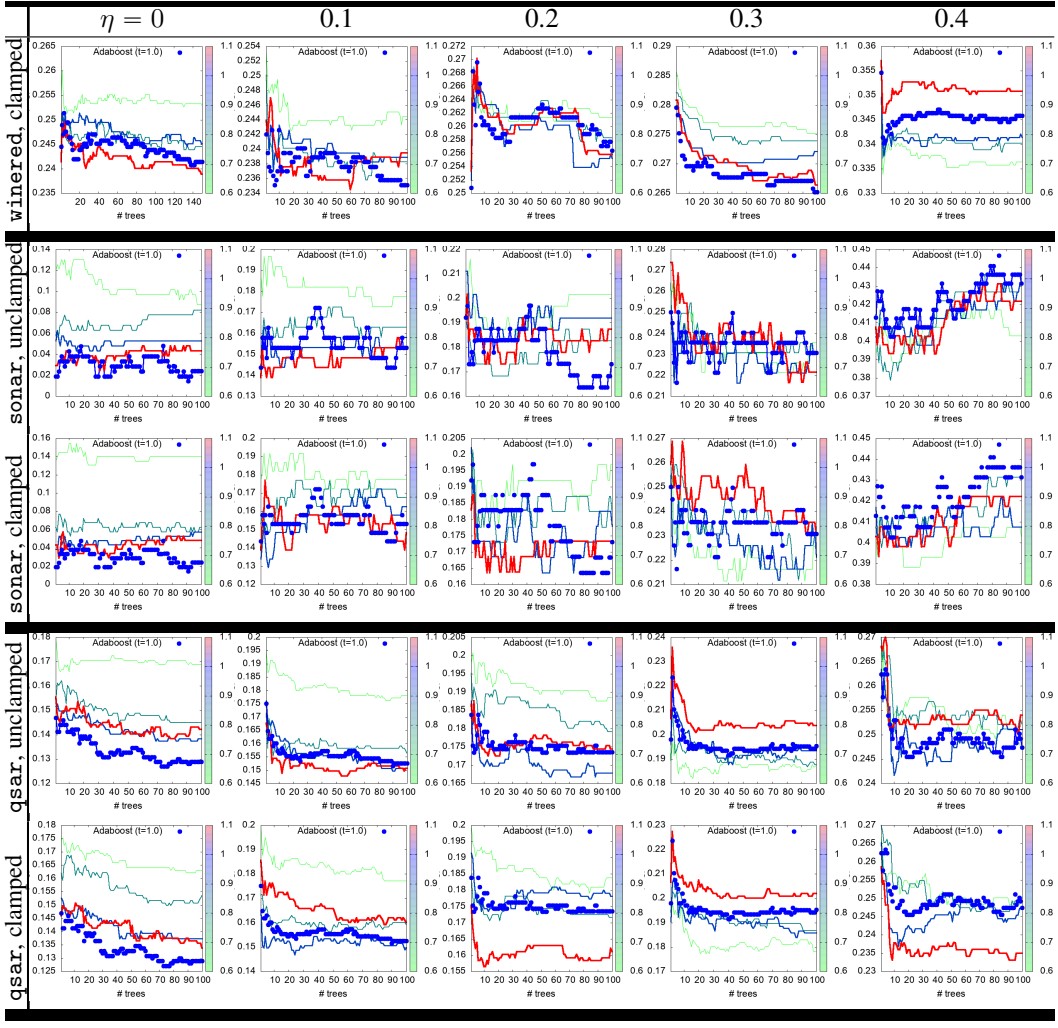

Table A16: Additional experiments with a larger number of trees and various levels of noise, on winered ($J = 150$ trees, $T = 2$ splits per tree) and sonar and qsar ($J = 100$, $T = 2$). Note that the range of noise levels tested is broader than for the other experiments.

