# OpenReview forum: "Boosting with Tempered Exponential Measures"
_NeurIPS.cc/2023/Conference — NeurIPS 2023 poster_

### Official Review · Reviewer_XQ4B · 2023-06-26

**Soundness:** 4 excellent
**Presentation:** 4 excellent
**Contribution:** 3 good
**Rating:** 7
**Confidence:** 4

**Summary:**

This work proposes a generalization of the popular ADABOOST algorithm based on the use of the t-logorithm/exponential. Their method is derived by replacing the standard relative entropy with the _tempered_ relative entropy (introduced in eq. 2), and solving a constrained optimization problem (eq. 3). This results in a solution (eq.4) which is a tempered generalization of ADABOOST's exponential update. Their approach recovers the standard ADABOOST updates at t->1, and maintains the exponential convergence rate of ADABOOST for values of 't' between 0 and 1. A new family of tempered losses is derived from the loss that t-ADABOOST minimizes, and experimental results using t-ADABOOST+trees show significant improvements can be gained by tuning 't'.

**Strengths:**

- This paper has both strong theoretical and experimental results, and show compelling improvements over baselines
- ADABOOST is a very popular and performant ensemble technique, and so this work has the potential to have significant applications for ML practitioners, and could easily be implemented in existing libraries (sklearn etc )
- The paper is well written and easy to follow, and their algorithm is accompanied with a comprehensive theoretical analysis

**Weaknesses:**

- The major limitation is that performance is highly sensitive to the choice of t, and currently it looks like the only way to choose t is to perform expensive hyperparameter sweeps. Further, their Table 1 (should that be "Figure 1"?) seems to imply there is no rhyme or reason to the datasets which perform better with large/medium/small t. Unless practitioners have a way of learning/tuning t it's unlikely this approach will be adopted widely in practice
- AFAICT, there are no ablation experiments testing the effect of the new losses, separate from the effect of the new exponential update. Which loss(es) were used in section 7? How do we know they work? How should a practitioner choose their loss?


## Four dimensions

**Originality:**
- Are the tasks or methods new?
  - Yes
- Is the work a novel combination of well-known techniques?
  - Yes, the work builds off developments in ADABOOST showing connections to exponential families and bregman divergences, and uses this to generalize ADABOOST by replacing exp/log with t-exp/t-log
- Is it clear how this work differs from previous contributions?
  - Yes, section 2 discusses this well.
- Is related work adequately cited?
  - Yes.

**Quality:**
- Is the submission technically sound?
  - Yes, the technical contributions seem sound.
- Are claims well supported (e.g., by theoretical analysis or experimental results)?
  - Theoretically yes, experimentally yes, with the exception of missing experiments evaluation the new losses
- Are the methods used appropriate?
  - Yes
- Is this a complete piece of work or work in progress?
  - Yes, modulo missing loss experiments
- Are the authors careful and honest about evaluating both the strengths and weaknesses of their work?
  - yes

**Clarity:**
- Is the submission clearly written?
  - Yes
- Is it well organized?
  - Yes
- Does it adequately inform the reader?
  - Yes
**Significance:**
- Are the results important?
  - If the authors provide a mechanism of choosing t, yes the results would be important.
- Are others (researchers or practitioners) likely to use the ideas or build on them?
  - Yes
- Does the submission address a difficult task in a better way than previous work?
  - Yes
- Does it advance the state of the art in a demonstrable way?
  - Yes
-  Does it provide unique data, unique conclusions about existing data, or a unique theoretical or experimental approach?
   - Yes


**Questions:**

- In https://arxiv.org/abs/2107.00745 the authors show t is a highly sensitive hyperparameter, and best results are obtained using tiny perturbations away from 1. This is due to numerical problems associated with using a log-t density instead of the more numerically-stable log-density. Did the authors see similar effects? Might their grid search of [0,0.2, 0.4, 0.6, 0.8, 0.9] be too coarse? Could significant gains be achieved with a finer grid search?
- If I'm a practitioner who is willing to do a grid search to find the best t, why wouldn't I spend that effort just tuning the hyperparameters of standard ADABOOST instead? If I take a hyperparameter-tuned ADABOOST, _then_ tune t, do I get significant gains compared to tuning t with untuned ADABOOST?
- With the introduction of the new losses in section 6, does that introduce a second t I need to tune, or should I use the same one?
- Does the second column of table 1 indicate clamping isn't effective?
- Do all/most values of t pick out the same "difficult" examples, or are the weights totally different for diff values of t?
- Is there any relationship between number of examples N and number of features D, and the best t? i.e do "tall+skinny" datasets get different optimal t's than 'short and fat' datasets, and might this offer tuning suggestions to practitioners? What about the depth of the weak classifier?
- ADABOOST can also be used for regression - would their approach work in this setting? A few comments about regression would be useful even if not feasible under the proposed approach.

**Limitations:**

There does not appear to be a "Limitations and Broader Impacts" statement in this work.

---

> ### Author Rebuttal · Authors · 2023-08-05
>
> We thank the reviewer for having evaluated our paper despite its heavy notational content.
>
> > The major limitation is that performance is highly sensitive to the choice of t,
>
> [XQ4B:A] We respectfully disagree with the argument: our theory says that regardless of the value of $t \in [0,1]$, the regime of performances do not degrade (both from the standpoint of the 0/1 loss, but also in terms of margins [4pmA:B][4pmA:C][4pmA:D]). Our experiments show that different datasets correspond to different values of “good” $t$s on test. Thus, there is an incentive to **not** stick with a single $t$ (e.g. AdaBoost).
>
> Now, more broadly – since the reviewer makes the comments from the angle of practical considerations –, ML is a field where heavy-hyperparametered algorithms abound: in deep learning, apart from architectural choices (use BatchNorm vs. LayerNorm, add/remove skip-connection, add/remove dropout, etc.), there are several hyperparameters to tune: starting with the optimizers, say, Adam, there are four parameters (lr, beta1, beta2, eps), plus the schedule for the learning rate (number of warm-up steps, decay type, etc.), weight decay parameter, etc.. Even in the field of tabular data learning, these are very common: the number of hyperparameters of XGBoost is 2-digits. In this big world of heavily tuned algorithms, adding one extra hyperparameter does not add too much complexity (compared to deep learning methods; even a ResNet-50 model has an order of two digits or more hyperparameters + plus many more architectural design choices). **This being said**, it would be nice to get “good guesses” for $t$ beforehand. *However*, it emerged after experiments suggested (**see attached pdf**) [6L9a:H], we in fact believe that the best way to approach the question could in fact be to **learn** $t$ during training and adapt it at *each* iteration, thus equivalently ** learning the loss**, a problem getting traction in ML.
>
> We also dispute the fact that there is “no rhyme or reason” in the results: in many cases, one could group the results for small $t$ and “big” $t$, with one group performing better than the other. That this picture changes among domains is not a downside: it is the *opposite*. It shows that the problem is worth solving *and* it is non-trivial. But it is out of the scope of our paper. We can even say it *has to be* out of the scope of our paper: almost all reviewers notice the already heavy load of material, which is probably to be made even heavier after some fruitful remarks [4pmA:B][4pmA:C][4pmA:D]. It would be a *tour-de-force* to fully explore *in addition* all dimensions of this problem. In fact, we claim it probably deserves its own paper (also considering the problem of learning the loss, see above) !
>
>
> > AFAICT, there are no ablation experiments testing the effect of the new losses
>
> [XQ4B:B] We do not understand this question: each value of $t$ gives rise to a different loss to optimize. So each time we fix a different $t$, the problem solved radically changes. Even more (Section 6), the full range of $t$ for the induction of decision trees covers the full range of known boosting rates !
>
> > How should a practitioner choose their loss?
>
> This is exactly the question / problem of choosing $t$ [XQ4B:B] ! We hope that this, the discussion in [XQ4B:A] and the new experiments (**see attached pdf**) [6L9a:H] shows the interest of our experiments and the fact that this problem deserves its devoted “iteration” (=paper).
>
> (questions)
>
> > In https://arxiv.org/abs/2107.00745 the authors [...] Did the authors see similar effects?
>
> Essentially, no. One reason is that we address a completely different problem.
>
> > Could significant gains be achieved with a finer grid search
>
> This is precisely relevant to [XQ4B:A].
>
> > why wouldn't I spend that effort just tuning the hyperparameters of standard ADABOOST instead?
>
> Because we add a new “robust” (=for which the theory behind stand still) dimension for which experiments – as the reviewer has already remarked – clearly show that there is value in exploring this dimension [XQ4B:A]. And depending on the package, the way some hyperparameters are fixed might just be recycled for a standard search for better $t$ than $t=1$ (AdaBoost). We also refer to the discussion [XQ4B:A].
>
> > should I use the same one?
>
> We use the same one.
>
> > Does the second column of table 1 indicate clamping isn't effective?
>
> Quite the opposite: it shows that it can be quite effective (for example, $t=0.9$ and **see attached pdf** [6L9a:H]) – with the additional benefit of doing computations for training / inference that remain in a prescribed “precision” interval, which could especially be relevant to specific applications [KBAC:A].
>
> > Do all/most values of t pick out the same "difficult" examples,
>
> Good question. From a purely theoretical standpoint, we would not expect this to be true, in particular for the comparison clamped / unclamped models.
>
> > Is there any relationship between number of examples N and number of features D, and the best t?
>
> There does not seem to be. We conjecture this has more to do with properties of the full domain itself.
>
> > ADABOOST can also be used for regression
>
> Good question, we conjecture it is possible, though some care-and-caution has to be applied, in particular for clamped models.

---

> > ### Comment · Reviewer_XQ4B · 2023-08-11
> > **Thank you for your response**
> >
> > Hi - I found your rebuttal convincing and appreciate the thoroughness of your response. None of the issues I raised were make-or-break, so I will keep my scores as they are. Looking forward to reading the final camera ready!

---

### Official Review · Reviewer_J18J · 2023-07-04

**Soundness:** 3 good
**Presentation:** 3 good
**Contribution:** 3 good
**Rating:** 6
**Confidence:** 4

**Summary:**


The paper introduces a generalization of the classic adaboost algorithm to apply to a family of certain exponential losses, called TEMs (tempered  exponential measures). They demonstrate the validity of their approach both theoretically  and empirically.




**Strengths:**

This technique allows their method to overcome numerical issues that typically arise when classic adaboost is employed.
Experiments show that tuning t can lead to significant improvements, compared to adaboost.
Moreover, the algorithm is simple and gives an interesting and practical generalization of adaboost.


**Weaknesses:**

My main issue here is the notational choices and somewhat unclear technical presentation.

Notation issues :
- Intuition regarding the used notation (in particular the terms in Eq. 4) would have been helpful, and also for the definitions of log_t and exp_t. Especially as these are key parts of the main algorithm which is very simple otherwise.
- Equation 5 is unclear - how is \mu defined? and the notation Z_t(\mu)  is confusing.
- What is Card? (Line 140) is that short for cardinality?
- Eq 14 - I’m assuming these are indicators returning +1/-1  ?


Because of the notation issues above, Theorem 2 is harder to parse. However, it does resemble in form to the standard bounds on adaboost - maybe it would be nice to emphasize that comparison more explicitly.
Also, consider giving more explanation of the meaning of theorem 2.
Maybe the paragraphs below it try to do that, but it was a little dense and technical.

I am also confused about Figure 1 - it is essentially plotting the convergence rate, the arrows indicate the case of t=0 and t=1, but is that something easily seen by the connection of the x-axis of rho to the value of t? Is it possible to also give a plot with an axis for values of t ?

Table 1 does not say what each line indicates (it did say on the appendix though) but still not sure what is the red line?

**Questions:**

Questions in the above comment

---

> ### Author Rebuttal · Authors · 2023-08-05
>
> We thank the reviewer for having evaluated our paper, despite its heavy technical nature.
>
> (weaknesses)
>
> > My main issue here is the notational choices and somewhat unclear technical presentation.
>
> [J18J:A] We apologize for the inconvenience, also probably due to several typos – fortunately spotted by the reviewers – that would have dampened the readability. As we explain in the general rebuttal, we believe there is a way forward to making it more readable. It is important as it has been asked that we consider additional theoretical results that will incur a few additional notations [4pmA:B][4pmA:C][4pmA:D].
>
> > Intuition regarding [...]
>
> [J18J:B] We believe this is one of the first times the tempered algebra (extending that over the reals) is used. To ease the reading, we propose to add in Section I of the supplement not just a primer on TEMs, but also a primer on this algebra and its properties.
>
> > Equation 5 is unclear [...]
>
> We used the same formalism as [11] (their eq. (1.5))
>
> > What is Card? (Line 140) is that short for cardinality?
> Correct. We will make this explicit.
>
> > Eq 14 - I’m assuming these are indicators returning +1/-1 ?
>
> No, this is the Iverson’s bracket (see Knuth in [12]), returning the truth value in ${0,1}$
>
> > Maybe the paragraphs below it try to do that, but it was a little dense and technical.
>
> [J18J:C] This is actually correct. Hopefully, with the additional page, we can put a bit of spacing in here to make it more readable
>
> > I am also confused about Figure 1 - it is essentially plotting the convergence rate
>
> No. The function plotted is the one in (12). The $x$-value is in fact $\rho_j$, which is in $[-1,1]$. The *color code* provides all different curves for all the different $t$s that are relevant. To simplify, the LHS of (12) is of the form $(K_t(\rho))^J$. Hence, if $K_t(\rho)$ is smaller than 1 (and the smaller it is), we have geometric convergence (and the faster it is). The plot shows that the curve for AdaBoost ($t=1$) is in fact the “highest” among all, and thus provides the “worst” guarantee of all, for $t\in [0,1]$. We could put $t$ as an axis but this would make the plot a 3D plot, not necessarily easily parsable at this size.
>
> > Table 1 does not say what [...]
>
> The red / thickest line is for $t=1.1$. We picked this color code because it is in fact the only value of $t$ for which the convergence is not guaranteed by our theory – but experiments obviously display that it is working as well !

---

> > ### Comment · Reviewer_J18J · 2023-08-22
> >
> > Thank you for your detailed response and clarifications.

---

### Official Review · Reviewer_4pmA · 2023-07-04

**Soundness:** 3 good
**Presentation:** 3 good
**Contribution:** 2 fair
**Rating:** 6
**Confidence:** 3

**Summary:**

The paper proposes a variant of AdaBoost algorithm based on a generalized exponential function parametrized by the temperature. The generalized exp function also induces a new criterion for splitting nodes of decision trees. The paper shows a training error bound of the AdaBoost variant. The experimental results show that the generalized AdaBoost often perform better than AdaBoost.

**Strengths:**

The strength of the contribution is the potential advantages of the proposed exp function. The exp function can derive new Bregman divergences and be applied to online learning or Boosting. In a practical sense, the new exp function and updates derived based on it seem more robust in numerical computation.

Another strength is a theoretical guarantee of its convergence rate of training error, which is the same as AdaBoost's, when given weak hypotheses with edge gamma. This iteration bound is optimal in the worst case (when the final hypothesis is a majority vote).

**Weaknesses:**

A crucial weakness of the paper is the lack of explanation about the generalization ability of proposed algorithms. Previous boosting algorithms, including AdaBoost, are motivated by their margin maximization properties. In fact, standard generalization bounds of weighted-voting classifiers depend on the margin over the sample.  Furthermore, there are boosting algorithms explicitly designed to optimize the soft margin optimization problem (e.g., SoftBoost or entropy regularized LPBoost). So far, the paper only considers the convergence rate of the training error.

So far, theoretical results do not improve previous ones, say, the convergence rate. So, the merit or theoretical advantages of the proposed function and the resulting boosting algorithm are unclear yet.


I read the rebuttal comments and the new analysis seems to resolve my concern.

**Questions:**

-Does the temperature control the smoothness of the function or resulting distributions over the sample? If so, please discuss the relationship between the previous boosting algorithms that keep distributions smooth (e.g., SmoothBoost by Servedio or AdaFlat by Gavinsly).




-

**Limitations:**

As raised above, the paper does not show any theoretical advantages of the proposed function. That is a huge drawback of the paper and thus the paper seems pre-matured.

---

> ### Author Rebuttal · Authors · 2023-08-04
>
> > A crucial weakness [...] generalization ability [...] margin maximization properties.
>
> [4pmA:A] we conjecture in L187 that similar rates of convergence (as the one we provide for the 0/1 loss) hold for margins as well; we also do not discuss generalization. We would like to point out also that margins are not discussed in the original AdaBoost paper [A] because this was not the focus of the paper. Historically, margins came later in the explanation as to why AdaBoost indeed works so well.
>
> While we initially thought that our paper was already dense enough to leave margins + generalization for a further step, the additional page of camera-ready might be used in part to address the reviewer’s concern. We have two good news in the direction they point at: one from the standpoint of margins, one on generalization. We apologize in advance for the use of Markdown to formally summarize those results (we were told that the 1 page pdf cannot be used for proofs).
>
> [4pmA:B] 1. Margins bounds. Instead of bounding $L_{0/1}(H)$ as in (12), we want to upperbound $E_i[f(y_iH(x_i)) \leq \theta]$ where $f(z)$ is increasing and typically in a bounded domain, say $[-1,1]$. [B] normalize their hypothesis space by the sum of leveraging coefficients. [C] prefer to pass the unnormalized classifier, with arbitrary real value, through the $\tanh$ function. A simple way forward for us is to generalize the approach of [C] to TEMs as well, which is in fact easy. Define the tempered hyperbolic tangent as $\tanh_t(z) = (1-\exp_t(-2z))/(1+\exp_t(-2z))$ and the margin of $H$ on example $(x, y)$ as:
>
> $\nu((x, y), H) = \tanh_t(yH(x)/2)$.
>
> Developing and reorganizing the predicate $[[\nu((x, y), H) \leq \theta]]$, it is the same as predicate
>
> $-yH(x) + \log_t\left(\frac{1+\theta}{1-\theta}\right) - (1-t) yH(x) \log_t\left(\frac{1+\theta}{1-\theta}\right) \geq 0$,
>
> Which, with the tempered algebra of [16] (our paper) is just stating $(-yH(x) \oplus_t \log_t\left(\frac{1+\theta}{1-\theta}\right)) \geq 0$. Since $[[z \geq 0]] \leq \exp_t^{2-t}(z), \forall t \in [0,1], \forall z\in \mathbb{R}$ ($[[.]]$ = Iverson’s bracket) and $\exp_t(u \oplus_t v) = \exp_t(u) \cdot \exp_t(v)$, we derive:
>
> $[[\nu((x,y),H) \leq \theta]] \leq \exp_t^{2-t}\left[ (-yH(x)) \oplus_t \log_t\left(\frac{1+\theta}{1-\theta}\right)\right] = \exp_t^{2-t}\left[ \log_t\left(\frac{1+\theta}{1-\theta}\right)\right] \cdot \exp_t^{2-t}(-yH(x))$,
>
> And so, after simplification, we get
>
> $[[\nu((x,y),H) \leq \theta]] \leq \left(\frac{1+\theta}{1-\theta}\right)^{2-t} \cdot \exp_t^{2-t}(-yH(x))$.
>
> We then branch directly to Section II.II.2.3, eq. 24, replacing $[[\mathrm{sign}(yH(x)) \neq y]]$ by $[[\nu((x,y),H) \leq \theta]]$, which yields in lieu of the (unnumbered) identity just before Lemma F,
>
> $\frac{1}{m} \cdot \sum_i [[\nu((x_i,y_i),H_J) \leq \theta]] \leq \left(\frac{1+\theta}{1-\theta}\right)^{2-t} \cdot \prod_{j=1}^{J}Z_{tj}^{2-t}$,
>
> and the rest of the proof of Theorem 2 remains unchanged. So, the left ineq. (12) (to save space) in our paper becomes the much more general inequality on **margins**:
>
> $\frac{1}{m} \cdot \sum_i [[\nu((x_i,y_i),H_J) \leq \theta]] \leq {\color{blue}\left(\frac{1+\theta}{1-\theta}\right)^{2-t}} \cdot\prod_{j=1}^J \tilde{Z}^{2-t}_{tj}$
>
> When $\theta = 0$, we recover (12) and if $t=1$, the ${\color{blue}\mbox{blue}}$ factor above recovers the one in Theorem 1 in [C]. First discovery, thanks to the reviewer:
>
> [4pmA:C] **$t$-AdaBoost is a margin maximization algorithm** (for any $\theta\in[-1,1), t\in[0,1]$)
>
> Drilling a bit reveals a perhaps more interesting phenomenon: when $\theta < 0$ (examples badly classified, eventually with large confidence), the blue factor $\left(\frac{1+\theta}{1-\theta}\right)^{2-t}$ can be substantially *smaller* than the same factor for $t = 1$ (this is the difference between $z$ and $z^2$ for $z\in [0,1]$), while when $\theta > 0$ (examples receiving the right class), the blue factor can this time be substantially *larger* than the same factor for $t = 1$. Hence, the analysis brings the interesting second discovery that
>
> [4pmA:D] **Fixing $t<1$ increases the “focus” of $t$-AdaBoost on increasing the margins of examples badly classified, compared to AdaBoost ($t=1$)**
>
> We certainly propose to use part of the camera-ready to state and prove [4pmA:C] and [4pmA:D], with due acknowledgements.
>
> [4pmA:E] 2. Generalization guarantees (simplified analysis, summarized). Suppose $H \in [-v,v]$, i.e. $|H|$  is bounded (e.g. we learn clamped classifiers). Then it is straightforward to see that the Lipschitz constant $L_t$ of the tempered exponential loss in $[-v,v]$ is $L_t = (2-t) \exp_t(v)$. If $v$ is sufficiently large, then $L_t < L_1$, $\forall t \in [0,1)$. Hence, from [D], the empirical minimization of the tempered exponential loss is better “aligned” with generalization for $t<1$ compared to AdaBoost. Note that if $v$ can be small, a more careful analysis is required, but could lead to nice characterizations of the appropriate *loss* to minimize for convergence rate *and* generalization.
>
> > So far, theoretical results do not improve previous ones [...] unclear yet.
>
> Even without [4pmA:C], [4pmA:D] and [4pmA:E], we strongly disagree with this statement. Just one example (space limit): it has been known for decades that AdaBoost can quickly run into numerical errors [13] because of its unbounded leveraging coefficients. We show that $t<1$ just gets rid of this problem, **and** at no cost convergence-wise.  Nor margin-wise [4pmA:C], [4pmA:D]. Nor generalization-wise [4pmA:E].
>
> > Does the temperature control [...] SmoothBoost [...] AdaFlat
>
> Briefly, $t<1$ allows to control the divergence of weights – equivalently, the smoothness parameter as in D. Gavinsky – (vs explicit in both SmoothBoost and AdaFlat)
>
> References:
>
> [A] Freund & Schapire, JCSS 55, 119-139, 1997.
> [B] Schapire, Freund, Bartlett & Lee, ICML 1997, 322-330.
> [C] Nock & Nielsen, ECAI 2006, 509-515.
> [D] Bartlett & Mendelson, JMLR 3, 463-482, 2002.

---

> > ### Comment · Reviewer_4pmA · 2023-08-12
> >
> > Thanks for the detailed reply to my concerns. The new part on margin error bound is convincing to me (but I am unsure if the new analysis might go beyond the rebuttal answer). I would appreciate if you could add the new analyses in the final version. I will raise my score.

---

> > > ### Author Response · Authors · 2023-08-13
> > > **Thank you**
> > >
> > > We thank very much the reviewer for the reply. We are committed to putting the margin analysis in the camera-ready, inclusive of comments -- in particular for the interesting behaviour [4pmA:D] --, using part of the additional space available.

---

### Official Review · Reviewer_tiN4 · 2023-07-25

**Soundness:** 3 good
**Presentation:** 3 good
**Contribution:** 3 good
**Rating:** 6
**Confidence:** 2

**Summary:**

This paper introduces a generalization of Ada Boost called “t-Ada Boost”. Boosting algorithms aggregate multiple weak classifiers into a strong classifier. Ada Boost is a well-established boosting algorithm.
t-Ada Boost generalizes Ada Boost by introducing an additional parameter $t$, which is a “tempering” parameter. When $t=1$, the algorithm becomes identical to Ada Boost. Thus, the main focus of the paper is to analyze whether or not any benefit can be gained by selecting a different value of $t$, besides $t=1$.
The paper provides a thorough theoretical and experimental analysis of t-AdaBoost, for various values of $t$. The theoretical analysis shows that different values of $t$ are viable to study, and bounds the convergence rate. The experimental analysis shows that, while $t=1$ works best for some data sets, and some values of $t$ seem equivalent to AdaBoost on other data sets, there do exist some data sets for which various values of $t$ perform better than Ada Boost. So, the overall analysis suggests that $t$ would be a good additional parameter to introduce on top of AdaBoost. For some data sets, tuning $t$ could lead to improvements on top of AdaBoost, so it’s worth consideration.


**Strengths:**

I think the paper is well-written, and conducts a thorough analysis of the proposed technique. It is built upon well-respected approaches.

**Weaknesses:**

The proposed method can have numerical instabilities, resulting from exponentiation, but the authors give reasonable consideration of this in their theory and experiments. This weakness is not very significant.

Also, see my experiment question below.


**Questions:**

-	Originally, I went looking for the full plots from the experiments, and did not see them in the supplement. I then realized that they occurred after the references. Perhaps add an additional entry in the “Table of Contents” would be appropriate, or else make sure the experiment plots are before the references.
-	Please include the code, or a link to the repository, along with the final version.
-	For some data sets, the plots see a high variability from changing the decision trees (see, for example, sonar, bottom). I know you mentioned that some of this might be due to overfitting. However, I also see that, in some cases, the error repeatedly rises and falls as the number of decision trees increases. This makes me wonder whether the variability here is something intrinsic. For example, if you rerun your experiments (perhaps with a very slight variation), do you get the same results? If not, then I think it is important to execute multiple runs and give a standard deviations, or other variance measure.


**Limitations:**

I don't think this paper has any negative societal impacts.

---

> ### Author Rebuttal · Authors · 2023-08-05
>
> (questions)
>
> > Originally, I went looking for the full plots from the experiments,
>
> Excellent suggestion ! We will oblige.
>
> > Please include the code,
>
> We commit to sharing all codes, inclusive of the plotting code
>
> >  If not, then I think it is important to execute multiple runs and give a standard deviations, or other variance measure.
>
> All our experiments are done using a 10-folds stratified cross-validation (L246). Thus, they include as well the standard deviations. We chose not to plot the deviation on the pictures to keep the pictures readable (we tried with, readability was very poor for many plots). **Hovever**, we emphasize that the summary results in Table 2 use a Student paired $t$-test to assess significance (Cf Table), and thus integrate the variability for comparisons.

---

> > ### Comment · Reviewer_tiN4 · 2023-08-11
> > **Thank you for the follow-up**
> >
> > I have read your rebuttal comments, and appreciate the clarifications.

---

### Official Review · Reviewer_KBAC · 2023-07-26

**Soundness:** 3 good
**Presentation:** 2 fair
**Contribution:** 3 good
**Rating:** 6
**Confidence:** 2

**Summary:**

The paper proposes a generalization of the ADABOOST algorithm using tempered exponential measures. To do this, they begin by introducing $log_t$ and $exp_t$ and the generalization of entropy. A first theorem is proposed to explain how to find the solution for the minimization of this entropy and an adaptation of the ADABOOST algorithm, called t-ADABOOST, is presented. Section 5 is dedicated to explaining the convergence of the t-ADABOOST algorithm and a theoretical study of its behavior. In section 6, the authors take the time to introduce tempered loss and the associated decision trees. The authors conclude with experiments and a discussion.

**Strengths:**

I think the paper is quite original in that the authors generalize ADABOOST using the functions $\log_t$ and $\exp_t$. In this way, they obtain an algorithm generalizing the former and extending the perspectives.
I was interested in the reflections on $\log_t$ and $\exp_t$ and how the introduction of these functions forced the authors to rethink the rest of the procedure.
The authors propose theorems that are consistent with our expectations as readers of these new notions, and the order seems logical.
Finally, the authors have chosen a complete path (new notions + theory + experiments + discussions), which makes the whole coherent in my opinion.


**Weaknesses:**

For me, a big weakness of the paper is the introduction of numerous notations that I didn't always find interesting (for example, I didn't understand the use of clamped sums) and which makes reading difficult. Certain notations, such as $t^*$, complicate reading in my opinion (because when $t$ and $t^*$ behave in opposite ways, you have to do some mental gymnastics to follow the reasoning). Finally, some notations are used differently: for example, $\textbf{q}$ is sometimes a vector (Definition 3.1) and sometimes a matrix (Algorithm 1), or $Z_t$ is sometimes a function in $\mu$ and sometimes not.
Similarly, the graphs are not intuitive for me. For example, why did the authors choose a color scale when there are only 7 values $t$ presented?
A last small weakness of this paper for me is the fact that the authors present "just" an extension of the ADABOOST algorithm but I do not have the impression to know if it will help or not to improve things (it is a point of discussion). I would have liked them to highlight at least one case where the algorithm actually worked around a weakness in ADABOOST.


**Questions:**

I think the paper would be easier to read if the number of notations were reduced (for example, $t^*$ is not compulsory in my opinion, as it's often $1/t^*$ that's used). I think it should be made clear whether $Z_t$ is a function or not. In the same vein, I think some equations lack rigor. For example, $\log_t$ and $\exp_t$ are introduced at the same time, even though their spaces of definition are different. On line 106, $\mu$ is not introduced either.
I didn't understand where clamped sum was used. Given the limited space available, I think there's a point in putting a paragraph on it. Could the authors explain?
What does the $\|\|\cdot\|\|_{1/t^*}$ norm represent? Is it just the Hölderian norm with $\alpha=1/t^*$? Or is it a new norm based on the new dissimilarity measure introduced?
I thought I read a few typos:
* Page 2, line 90: [11, 6, 17] is not in increasing order.
* Page 7, line 246: Table A1 or Section III1 instead of Section A1?
* Page 8, Table 1: isn't it a figure?

**Limitations:**

I didn't see any limitations that weren't addressed by the authors. I appreciated their honesty.

---

> ### Author Rebuttal · Authors · 2023-08-05
>
> First, we thank the reader for having evaluated our paper despite its heavy notational nature.
>
> (weaknesses)
>
> > For me, a big weakness of the paper is the introduction of numerous notations that I didn't always find interesting
>
> [KBAC:A] We would like to emphasize that it has been asked that we consider additional theoretical results that will add additional notations [4pmA:B][4pmA:C][4pmA:D]. We do believe there is value in those additional results, and we believe that what the proposal we make (general rebuttal) to better present our notations will be of great help to understand the paper. We also believe that some hiccups in reading were in fact created by typos that the reviewers have spotted; removing them will substantially contribute to easing the reading.
> For clamped summations, we in fact do believe that such classifiers could be of substantial use, in particular when training or inference is done with *reduced computational power* (either because of machines, e.g. ML at the edge, or because of compute constraints, e.g. MPC / secure multiparty computation). It is important to realize that clamping / clipping a value -- as it is usually done -- still requires *storing the full value* before “simplifying” it (or you lose arbitrary numerical precision, which can be damaging). In our case, *all steps in the computation of the output can be performed with the desired complexity* and do not alter the guarantees.
>
> > q  is sometimes a vector (Definition 3.1) and sometimes a matrix
>
> No, $q$ is always a vector. The first index in the notation in Algorithm 1 is the iteration number, explicit in Step 2.
>
> > $Z_t$ is sometimes a function in $\mu$ and sometimes not
>
> No. Dependences may be implicitly noted to lighten the text. $Z$ is the normalization coefficient, which *de facto* is always a function of the leveraging coefficients (and their additional parameters, eventually). This is a consequence of (4).
>
> > For example, why did the authors choose a color scale when there are only 7 values  presented?
>
> Simply because we wanted to give the reader different ways to visually compare the curves, one using the thickness of the curves, one using the colors of the curves. By this choice, we hope to be more inclusive with respect to the readers of our paper.
>
> (questions)
>
> > I think the paper would be easier to read if the number of notations were reduced
>
> We hope the comments above clarify some proposed changes on $Z$.
>
> > for example, $t^*$  is not compulsory in my opinion
>
> This notation was introduced in [3]. We agree with the reviewer; however, we believe removing the $t^*$ notation (and substituting the value $\frac{1}{2-t}$) will eventually make the expressions longer and impair the readability of the derivations.
>
> > I think some equations lack rigor. For example, $\log_t$ and $\exp_t$ are introduced at the same time, even though their spaces of definition are different
>
> $\log_t$ and $\exp_t$ are indeed inverse functions introduced by Naudts (2011), as generalizations of standard $\log$ and $\exp$. If the reviewer is referring to their respective domains, we can make further clarifications in the text. Did the reviewer also want us to change the width of the equations ? (this would affect the height as well and would decrease the readability of our paper)
>
> >  $\mu$ is not introduced either.
>
> $\mu$ is just a real number. We can put $\mu \in \mathbb{R}$.
>
> >  I didn't understand where clamped sum was used
>
> As explained in (9), clamped sums are used to define models.
>
> > What does the $\|.\|_{1/t^*}$  norm represent?
>
> The $L_p$ norm with $p = 1/t^*$. Standard notation.
>
> We acknowledge the typos found and will make the modifications. Thanks !
>
> (limitations)
>
> > I didn't see any limitations[...]. I appreciated their honesty.
>
> We would like to very much thank the reviewer for this comment.
>
> (Referene)
> Naudts, J. (2011). Generalized thermostatistics. Springer

---

> > ### Comment · Reviewer_KBAC · 2023-08-19
> > **I have always some questions**
> >
> > I thank the authors for clarifying the text. I hope it is more understandable by the uninitiated.
> > > No, $q$ is always a vector. The first index in the notation in Algorithm 1 is the iteration number, explicit in Step 2.
> >
> > Ok, thanks. As $q$ is a vector, I think it would be better to use the $q^(1)$ notation for iteration.
> > > No. Dependences may be implicitly noted to lighten the text. $Z$ is the normalization coefficient, which de facto is always a function of the leveraging coefficients
> >
> > Sorry, I think my comment was misunderstood. In math, if $f$ is a function, $f(x)$ is a scalar. My problem is your choice to consider $Z_t$ to be a scalar when $Z_t(\mu)$ is the scalar. I understand that it is painful to write it every time but I find it more precise to add it.
> > > (and their additional parameters, eventually).
> >
> > If you say that there may be other parameters, I think it is all the more important to add them.
> > > This notation was introduced in [3]. We agree with the reviewer; however, we believe removing the $t^*$ notation (and substituting the value $\frac{1}{2-t}$) will eventually make the expressions longer and impair the readability of the derivations.
> >
> > We agree on the advantages/problems of both choices, and whether to take $t^*$ or $\frac{1}{2-t}$ is obviously just a matter of point of view. Mine remains that since you're talking about $t$ convergence, it's clearer for the reader to understand what's going on if they see $\frac{1}{2-t}$ directly rather than $t^*$. Alternatively, you could put convergences directly in $t^*$ to avoid any intellectual gymnastics.
> > > If the reviewer is referring to their respective domains, we can make further clarifications in the text. Did the reviewer also want us to change the width of the equations ?
> >
> > My remark was simply to remind you that $\log_t$ was defined on $\mathbb{R}^+_{\star}$ and $\exp_t$ on $\mathbb{R}$. Once again, I understand that, if there is no clarification, the first idea is to copy the definition sets of $\log$ and $\exp$. But since these are extensions, what proof do we have that the set isn't different? A bit like the extension of the logarithm to the imaginary, which can be defined almost anywhere.
> > > We can put $\mu\in\mathbb{R}$.
> >
> > Thanks.
> > > As explained in (9), clamped sums are used to define models.
> >
> > Sorry, my question was *where $H_J^{(\delta)}$ was used?* I must have had a moment's inattention because it was line 145. Sorry for my question.
> > > The $L_p$ norm with $p=1/t^*$.
> >
> > Thank you for your reply. I don't remember why, on first reading, it bothered me. Sorry about that.

---

### Official Review · Reviewer_6L9a · 2023-07-31

**Soundness:** 3 good
**Presentation:** 3 good
**Contribution:** 3 good
**Rating:** 6
**Confidence:** 3

**Summary:**

The paper presents an estension of ADABoost algorithm for binary classification, called t-ADABoost, by modifying the weight optimisation under simplex constraint formulation of the original algorithm. The generalised formulation involves optimisation of modified Bregman divergence between new and old weights under a so called co-simplex constraint (i.e. a sum constraint on the power of the weights). The paper establishes that these new optimisation problems amount to a one parameter optimisation problem in a way similar to ADABoost, then that the resulting construction of a strong learner will, under a generalised assumption, will have an empirical error which decreases exponentially quick.

The paper complements their results by experiments comparing the behavior of the t-ADABoost algorithm for decision tree using different t values. The results show that depending on the data and presence of noise, the relationship between t value and test error can change.

**Strengths:**

The paper is well introduced and describes an original extension of ADABoost algorithm, by incorporating the notion of tempered exponential measures which was found to improve robustness in clustering.

The authors give sufficient evidence that the extended algorithm is able to decrease the empirical risk as at least the same rate as the original ADABoost algorithm  (Theorem 2 and discussion following) and that the use of an oracle t-ADABoost algorithm can reduce the test error compared to the initial ADABoost algorithm (Table 1). This, as the authors note, imply that while work remains to be done in order to learn the optimal t for each new dataset, this generalisation has its use cases.



**Weaknesses:**

Part of section 5 and Algorithm 1 are confusing. Notably, in the description of Algorithm 1, the beginning of Step 2.2 starts with what seems to be a typo (I believe $\mu_{ji} = y_i h_j(x_i)$ should read $ u_{ji} = y_i h_j(x_i)$, then neither the coefficients $\nu_j$ and $\alpha_j$ are defined until Theorem 2. The wording "choose leveraging coefficient" seems to imply that the choice can be arbitrary, while, considering that Algorithm 1 is described just after Theorem 1., one could infer that the coefficient $\nu_j$ is the minimizer of equation (5).
In equation 11, the quantity $m^{1-t^*}$ is not introduced anywhere.  The next mention is in line 180, where it is stated that it can be discarded in the unclamped case, but that it does play a role in equation 12, where the quantity does not appear (while $1 + m^\dag {q_j^\dag}^{2-t}}$ appears and could play the "dampening part" mentionned later on).

Parts of section 6 could be clarified. There is some confusion between the notations $L$ and $\underline{L}$ (line 200, $\mathbb{E}_\lambda[\underline{L}(p_\lambda)$, but $\mathbb{E}_\lambda[L(p_\lambda)]$ line 219, which in itself does not make sense as $L$ takes 2 arguments. Moreover in equation (16), from equation 200 implies that the DT is trained from the Bayes risk, and as such, reverse engineering should start by computing the Bayes risk rather than CPE loss).

**Questions:**

The algorithms presented here works for binary classification, as is the case for the initial ADABoost algorithm. ADABoost has been extended to multiclass settings. Could such an extension also work for t-ADABoost?

The interpretation of theorem 2 relies (line 161) on the assumption that $\lvert \rho_j\rvert \geq \gamma$. If $m_j^\dag > 0$, the value of $\rho_j$ depends on $t$ and therefore the assumption above could be false for certain $t$. Is there any insight on how this impacts the exponential decay? Moreover, even in the case where $m_j^\dag = 0$, considering that $\rho_j = \frac{1}{R_j}\sum_{i\in[m]} q_{ji}, y_i h_j(x_i)$  and that $q_{ji}$ no longer need sum to $1$ but rather should belong to a co-simplex, the assumption that $\lvert\rho\rvert >\gamma$ does not have the same implication depending on the value on $t$. Could you comment on whether this is a stronger requirement in the case where $t<1$.

Are there any overfitting issues related to t-ADABoost, notably when $t \ll 1$ or when $t > 1$? Could these be investigated by allowing the number of trees grown to be larger than 20 in the experiments?

In section 6, it is mentionned that reference 10 note that Matusita's loss implies a near optimal boosting rate while the empriical risk gives the worst possible guarantee. Could you specify where these two results are stated in the reference?

**Limitations:**

The current form of the algorithm seems to be usable only the case of binary classifiation, while the original ADABoost has been extended to multiclass settings. It is unclear whether this restriction can be lifted efficiently.

The authors clearly mention that the current work should be complemented by further insights on the selection of $t$, which is coherent with their experiment results (where the optimal $t$ value depends on the dataset and noise).

---

> ### Author Rebuttal · Authors · 2023-08-05
>
> First, we would like to thank the reviewer for reading our paper and noticing the typos mentioned.
>
> (weaknesses section)
>
> > Part of section 5 and Algorithm 1 are confusing
>
> [6L9a:A] We sincerely apologize for the confusion, resulting from our choice of organization, and unwanted typos. Correct for $\mu_{ji}$.
>
> > The wording "choose leveraging coefficient" seems to imply that the choice can be arbitrary
>
> [6L9a:B] We followed the convention for AdaBoost [28] (our paper). We can do otherwise
>
> >  could infer that the coefficient $\nu_j$  is the minimizer
>
> [6L9a:C] (we assume it is $\mu_j$) It is a minimizer of an upperbound of (5) with the “secant trick” (II.II.2.4 supplement)
>
> > the quantity $m^{1-t^*}$
>
> [6L9a:D] $m$ is the number of training examples, $t^* = 1/(2-t)$. This is just a factor (the same for all leveraging coefficients) that makes the formal analysis “simple”. It logically disappears from (12) because it is part of $H$.
>
> > Parts of section 6 could be clarified
>
> [6L9a:E] As we state in L219, the simplification ends up with a loss of the form $E_\lambda[L(p_\lambda)]$. Here, $L$ may just be any function from $[0,1]$ to $\mathbb{R}$. It turns out that, in our case, it ends up being a very particular function with the key property of being proper – and thus eliciting Bayes prediction as its minimizer – for any $t\in [-\infty,2)$, a function usually noted $\underline{L}$ [22,23] (our paper). We can make this more explicit.
>
> Note: we insist on the fact that in a field (ML) where we are used to see a plethora of loss functions, such an invariant is quite exceptional because one can safely slide $t$ in a huge range without ever breaking up properness *and* guaranteeing boosting rates in the complete spectrum of known rates (and even unknown rates, for $t\in(1,2)$ (L243). We know of no other parameterized loss with such a property.
>
> (Questions section)
>
> > The algorithms presented here works for binary classification... Could such an extension [to the multiclass setting] also work for t-ADABoost?
>
> [6L9a:F] Most certainly, following the blueprint of [28], Section 6.
>
> > The interpretation of theorem 2 relies (line 161) on the assumption
>
> Excellent questions ! First, the theoretical analysis covers all cases, including $m_j^\dagger > 0$. In our experiments, we always noted $m_j^\dagger = 0$ so the first question seems to be essentially relevant to theory. The question(s) led us to reconsider why we observed this and in fact there is a simple explanation as to why $m_j^\dagger > 0$ would be very rare *if* the weak learner is not “that” weak (which is our case, with decision trees): one needs to consider Lemma J (supplement) together with (12). Let $Q_j = 1 + m_j^\dagger (q_j^{\dagger})^{2-t}$ and $\tilde{\rho}_j = \rho_j \cdot Q_j$. Note that $\tilde{\rho}_j$ is of the form $\beta \cdot E_{p_j}[yh]$ where $p_j$ lives on the *simplex*, and $|yh| \leq 1$, $\beta \leq 1$. Using Lemma J with (12), to keep geometric convergence, it is sufficient that
>
> $Q_j \log Q_j \leq  \tilde{\rho}_j^2 / (2t^*) $
>
> Since $q_j^{\dagger}$ is homogeneous to a tempered weight, one would expect in general $m_j^\dagger (q_j^{\dagger})^{2-t} \ll 1$, so using $Q_j \log Q_j \sim_1 -1 + Q_j$, one gets the sufficient condition
>
> $m_j^\dagger (q_j^{\dagger})^{2-t} \leq \tilde{\rho}_j^2 / (2t^*)$
>
> Note that for $t=0$, $2t^* = 1$, so one roughly gets that to keep geometric convergence, one needs $m_j^\dagger (q_j^{\dagger})^{2-t} = O(\tilde{\rho}_j^2)$. What does that mean ?
>
> 1. If it is not violated, then we have geometric convergence
> 2. If it is, then a large number of training examples have $q_{ji} = 0$, which means that they are receiving *the right class with large margin* [4pmA:C], [4pmA:D]. In this case, breaking geometric convergence is not an issue: we already have a very good ensemble !
>
> [6L9a:G] We propose to put this analysis, which we believe is enlightening, at least in the supplement.
>
> > Are there any overfitting issues related to t-ADABoost,
>
> [6L9a:H] Excellent question. We have performed additional experiments on learning ensembles with a larger number of trees ($J$), each of them being smaller ($T$) *and* tested domains with a larger amount of training noise (because noise affects just training, it could induce overfitting by models “focusing” more on fitting training data or the noise patterns, at the expense of the full domain). We crammed a few in **the attached .pdf**, selected for the topic raised (we would be in a position to present all results in the supplement of the camera-ready). What emerges:
>
> 1. Overfitting *can* happen (winered $\eta = 0.2, 0.4$, sonar $\eta = 0.4$) but affects very differently the algorithm at different $t$ values and yields very substantial differences (by several % points). Overall, this displays that tuning $t$ can also have the purpose of handling overfitting.
>
> 2. Clamped models can be better at resisting overfitting (qsar, all $\eta > 0$). Strong incentive to train clamped models as well.
>
> 3. Some plots (sonar $\eta > 0.1$) suggest the idea that more than just tuning $t$ beforehand, good strategies could in fact **learn** $t$ and adapt it **during training** (so each iteration $j\in J$ would use a specific $t_j$). We suggest adding this in conclusion.
>
> > In section 6, it is mentionned that reference 10 note that Matusita's loss
>
> [6L9a:I] Optimality of Matusita = paragraph following Theorem 1 in the JCSS version (open access). Empirical risk = worst: Fig 2 + Fig 5 (+legends). Note: [10] prove the optimality of Matusita’s from an information-theoretic standpoint. It is also shown from a computational complexity standpoint in [A]
>
> [A] Nock & Nielsen, “On domain-partitioning induction criteria: worst-case bounds for the worst-case based” (TCS 321, pp 371-382, 2004).

---

> > ### Comment · Reviewer_6L9a · 2023-08-18
> >
> > I have read the authors' answer, and thank them for their clarification.
> >
> > [6L9a:A], [6L9a:D], [6L9a:F], [6L9a:H], [6L9a:I] are satisfactory explained/taken into account.
> >
> > For [6L9a:B] and [6L9a:C]: First of all, you're right, $\nu_j$ in my comment should be read $\mu_j$. My comment did not concern the rationale for the choice of $\mu_j$, but was only concerned with the presentation of the algorithm: if reading the paper linearly, the first time $\mu_j$ is mentionned, it is not yet defined. Since it is defined a page later, it could be confusing. I would advise mentionning in the Algorithm caption that the different values involved in the algorithm are chosen from theorem 2, only to help readability.
> >
> > [6L9a:E] is cleared up, although I would still advise using a different letter than L in line 219 (I've only just noticed that the letter was not in italic, and as the notation L in italic stands for a function of 2 arguments as defined in equation 15, this can be confusing).
> >
> > I've read the development on [6L9a:F], though I'm unsure to understand it thoroughly.
> >
> > The authors conducted thorough analysis to assess resistance to overfitting in [6L9a:H]. As the authors note, experimental results suggest than the choice of $t$ and the use of clamped models can impact overfitting risks. This increases the impact of the extension.
> >
> >
> > All in all, the authors satisfactorily answered all remarks. I'm inclined to increase the rating from 6 to 7, mostly due to the impact the generalised Adaboost procedure could have on overfitting risks.

---

> > > ### Author Response · Authors · 2023-08-19
> > > **Thanks for the last comments & inclination to increase score (6 -> 7)**
> > >
> > > We thank the reviewer for their last comments and reported impact on changing their score.
> > >
> > > The reviewer mention inclination on changing rating from 6 to 7, though we are not sure it has been actioned in OpenReview.
> > >
> > > With regards,
> > > the authors.

---

### Author Rebuttal · Authors · 2023-08-07

We would like to thank all six reviewers for their work and appreciate the global positive tone of reviews given the notation-heavy nature of our paper. To ease the cross-search among the pieces of our rebuttal, we have put tokens of the form **[Reviewer-Id:Letter]** in rebuttals, making it easily to search for cross-references among rebuttals. Each rebuttal proceeds by quoting the review and replying, in the order of the review’s comments. References shown like [number] refer to our paper’s bibliography. References shown like [letter] refer to references put in each rebuttal.

This is a general rebuttal, summarizing the major changes, all of which could be easily done given the additional page in the camera-ready.

## From the standpoint of its theoretical content

We note that our paper *might* see an increase in notations after new results that we got for a rebuttal on margins and generalization **[4pmA:B][4pmA:C][4pmA:D][4pmA:E]**. We believe however that reviewers have done a fantastic job of typo-spotting and that the final version of our paper will surely gain in readability. We apologize however for having used the keen eye of six reviewers to find those.

To ease reading our paper, we propose

1. To correct all typos (we believe they contributed to hiccups in reading)
2. To put in the camera-ready before Section I in the supplement a table of notations, grouped by topics (models, TEMs, properness, etc.), eventually by paper Section, to help the reading of the paper
3. To strengthen part I in the supplement to include not just a primer on TEMs but also on the tempered algebra of [16]

## From the standpoint of its experimental content

Experiments vs overfitting suggested by **[6L9a:H]** have been put in **the attached .pdf**, and present an interesting direction which we believe could present value, briefly mentioned in the main file and then put in the supplement.

---

### Decision · Program_Chairs · 2023-09-21

**Decision:**

Accept (poster)

**Comment:**

This meta review is based on the reviews, the authors rebuttal and the discussions with the reviewers, discussions with the SAC, and ultimately my own judgement on the paper. There was a consensus that the paper contributes sound and interesting contributions. I feel this work deserves to be featured at NeurIPS and will attract interest from the community. I would like to personally invite the authors to carefully revise their manuscript to take into account the remarks and suggestions made by reviewers. Congratulations!